# Base-pair resolution analysis of the effect of supercoiling on DNA flexibility and major groove recognition by triplex-forming oligonucleotides

Alice L. B. Pyne [1,2,13✉], Agnes Noy [3,13✉], Kavit H. S. Main[2,4], Victor Velasco-Berrelleza [3], Michael M. Piperakis[5,6], Lesley A. Mitchenall[5], Fiorella M. Cugliandolo[5,7], Joseph G. Beton[2,8], Clare E. M. Stevenson [5], Bart W. Hoogenboom [2,9], Andrew D. Bates[10], Anthony Maxwell [5] & Sarah A. Harris [11,12✉]

In the cell, DNA is arranged into highly-organised and topologically-constrained (supercoiled) structures. It remains unclear how this supercoiling affects the detailed double-helical structure of DNA, largely because of limitations in spatial resolution of the available biophysical tools. Here, we overcome these limitations, by a combination of atomic force microscopy (AFM) and atomistic molecular dynamics (MD) simulations, to resolve structures of negatively-supercoiled DNA minicircles at base-pair resolution. We observe that negative superhelical stress induces local variation in the canonical B-form DNA structure by introducing kinks and defects that affect global minicircle structure and flexibility. We probe how these local and global conformational changes affect DNA interactions through the binding of triplex-forming oligonucleotides to DNA minicircles. We show that the energetics of triplex formation is governed by a delicate balance between electrostatics and bonding interactions. Our results provide mechanistic insight into how DNA supercoiling can affect molecular recognition, that may have broader implications for DNA interactions with other molecular species.

[1] Department of Materials Science and Engineering, University of Sheffield, Sheffield, UK. [2] London Centre for Nanotechnology, University College London, London, UK. [3] Department of Physics, Biological Physical Sciences Institute, University of York, York, UK. [4] UCL Cancer Institute, University College London, London, UK. [5] Department of Biological Chemistry, John Innes Centre, Norwich, UK. [6] Department of Chemistry, University of Reading, Whiteknights, Reading, UK. [7] Department of Pathology, Division of Immunology, University of Cambridge, Cambridge, UK. [8] Department of Crystallography, Institute of Structural and Molecular Biology, Birkbeck, University of London, London, UK. [9] Department of Physics and Astronomy, University College London, London, UK. [10] Institute of Integrative Biology, University of Liverpool, Liverpool, UK. [11] School of Physics and Astronomy, University of Leeds, Leeds, UK. [12] Astbury Centre for Structural Molecular Biology, University of Leeds, Leeds, UK. [13] These authors contributed equally: Alice L. B. Pyne, Agnes Noy. ✉email: a.l.pyne@sheffield.ac.uk; agnes.noy@york.ac.uk; s.a.harris@leeds.ac.uk

Genomic DNA is often subjected to torsional stress, which can both over- and under-wind the DNA double helix[1–3]. Negative superhelical stress results from a reduction in the number of links (Lk) between the two strands of a closed-circular DNA (a negative $\Delta$Lk). The conformational response to this stress is called negative supercoiling, partitioned between untwisting of the helix (change in twist; Tw) and a coiling deformation of the DNA backbone (writhe; Wr)[1–4]. In prokaryotes, genomic DNA has an average density of supercoiling, $\sigma$ ($\Delta$Lk/original Lk) of $\sim -0.06$[5]. Supercoiling operates synergistically with nuclear-associated proteins to regulate bacterial gene expression[6]. In eukaryotes, supercoiling generated by transcription is implicated in the regulation of oncogenes such as c-Myc[7]. It plays a fundamental role in the formation and stability of looped DNA structures[8] and DNA R-loops[9], and influences the placement of RNA guide sequences by the CRISPR-Cas9 gene editing toolkit[10]. The supercoiling-induced structural changes that modulate these DNA functions present a challenge for traditional structural methods that can provide atomistic resolution, that is, X-ray crystallography[11] and nuclear magnetic resonance[12], because of the diverse conformational landscape of supercoiled DNA[13].

As part of its role in regulating transcription, replication and chromosomal segregation[14], supercoiling has been proposed to play a role in the specificity of DNA-binding ligands, including major groove binders such as triplex-forming oligonucleotides (TFOs)[15,16]. TFOs target specific DNA sequences, forming a triplex of the single-stranded TFO and the target duplex DNA[17]. The target specificity of TFOs combined with their ability to suppress gene expression has driven their development as anti-cancer agents. TFOs provide an exemplary model system for studying the twist–writhe balance in supercoiling dependent DNA recognition. Triplex formation requires the DNA to be locally under-twisted to accommodate the third strand, and TFOs form intimate interactions with a relatively long span of DNA (16 bp) compared to typical DNA-binding domains in proteins (between 4 and 10 bp)[18].

Here, we combine high-resolution atomic force microscopy (AFM) with molecular dynamics (MD) simulations to reveal how supercoiling affects global and local DNA conformation, structure and dynamics in DNA minicircles of length 250–340 bp. These minicircles are small enough to be simulated at the atomistic level by MD[13,19] and to be visualised at high (double-helix) resolution by AFM experiments in solution[20–22]. Minicircles are also representative of looped DNA at plectoneme tips[23] and small extrachromosomal circular DNAs, which have tissue-specific populations and sequence profiles in human cells[24–26]. The DNA minicircles in this study incorporate a TFO-binding sequence, to assess how the interplay of electrostatic and base-stacking energies determines the formation of triplex structures in supercoiled DNA.

## Results

### High-resolution AFM and MD reveal conformational diversity in supercoiled DNA minicircles. 
Figure 1 shows the structure of negatively supercoiled DNA minicircles as viewed by high-resolution AFM and simulated by atomistic MD. High-resolution AFM images recorded in aqueous solution show DNA minicircles, isolated with native levels of supercoiling, in a range of conformations with sufficient resolution to resolve the two oligonucleotide strands of the double helix. For the 251 bp minicircle, this allowed determination of the linking number, Lk = 24 ± 1 from direct measurements of twist (24 ± 1 turns) and writhe (≤1). The measured twist corresponds to a helical repeat of 10.5 ± 0.5 bp, consistent with canonical B-form DNA[1]. For each conformation of the surface-bound minicircles found by AFM

(Fig. 1a–d), it was possible to find MD-generated conformers with a close resemblance in global structure (Fig. 1e) (see 'Methods' for details). The deviation from planarity of the minicircles was calculated to be <15% on average (Supplementary Fig. 1 and Supplementary Videos 1–5), which is advantageous to structural determination by AFM, because distortions resulting from surface immobilisation are minimal for planar molecules. Atomistic models of supercoiled DNA minicircles have been shown to be consistent with cryo-electron tomography density maps[13], which provide sufficient resolution to capture the overall shape of the minicircles, but not their helical structure. The variation in structures observed in Fig. 1a–e is attributed to thermal fluctuations within supercoiled DNA, with time-resolved AFM (Fig. 1f) demonstrating that dynamic behaviour can occur in these molecules on the order of minutes, even when tethered to a surface. These fluctuations could be in part induced by the energy imparted by the tip during AFM imaging, which allows the molecule to explore its energy landscape even while tethered to a surface. Similar dynamics were observed in MD simulations of the 339 minicircle ($\Delta$Lk = −1) in a continuum representation of the solvent, albeit at a much faster (picosecond) rate (Fig. 1g and Supplementary Videos 6 and 7). Experimental measurements have shown that adsorption to a surface for AFM slows dynamics[27,28], and in silico, the absence of friction with water molecules accelerates conformational dynamics[13,29]. The selected 2D projections of MD conformers that we compare to the AFM images occur in a different chronological order in the simulations due to the random statistical nature of thermal fluctuations (Supplementary Fig. 2).

### Negative supercoiling induces defects in DNA minicircles. 
AFM not only provides resolution sufficient to observe the DNA helical repeat but most critically achieves this without the need for ensemble averaging. This uniquely permits us to observe heterogeneous structural perturbations, for example, individual DNA defects, that occur due to superhelical stress imposed on the minicircle. By combining AFM and in silico measurements of DNA minicircle topoisomers with increasing levels of super-coiling (Fig. 2), we were able to observe the effect of negative supercoiling on the structure and mechanics of DNA with Ångström resolution. We observed no defects in the structure of the relaxed topoisomer, which maintains a B-form structure throughout the molecule. However, in negatively supercoiled DNA, defects were observed both by AFM (Fig. 1a–d, red triangles) and atomistic MD simulations (Fig. 2a, red triangles). We observed the onset of defects in negatively supercoiled minicircles of $\Delta$Lk = −1 onwards ($\sigma \approx -0.03$); across all in silico topoisomers, seven out of the ten defects observed are denaturation bubbles, where two or more base pairs are flipped out of the duplex (Fig. 2a, insets and Supplementary Fig. 3). This results in flexible hinges that can accommodate a 180° turn within a single helical turn, radically altering the range of conformations the DNA can adopt. We also observed type I kinks[30] in topoisomers −1 and −3 (in which a single base pair presents a strong bend, breaking hydrogen bonds and stacking), and a type II kink[30] within topoisomer −3 (in which hydrogen bonds of two consecutive base pairs are broken and bases are stacked on their 5′ neighbours) (Fig. 2a, insets). Equivalent bending and supercoiling-induced deformations have been reported in smaller minicircles (between around 60 and 100 bp) by MD simulations[30–32], by cryo-electron microscopy[33,34] and by biochemical analysis using enzymatic probes that selectively digest single-stranded DNA regions[35]. Based on insight from atomistic MD simulations[30], type I kinks and more severe disruptions have been associated with slow and fast enzymatic digestion, respectively. In

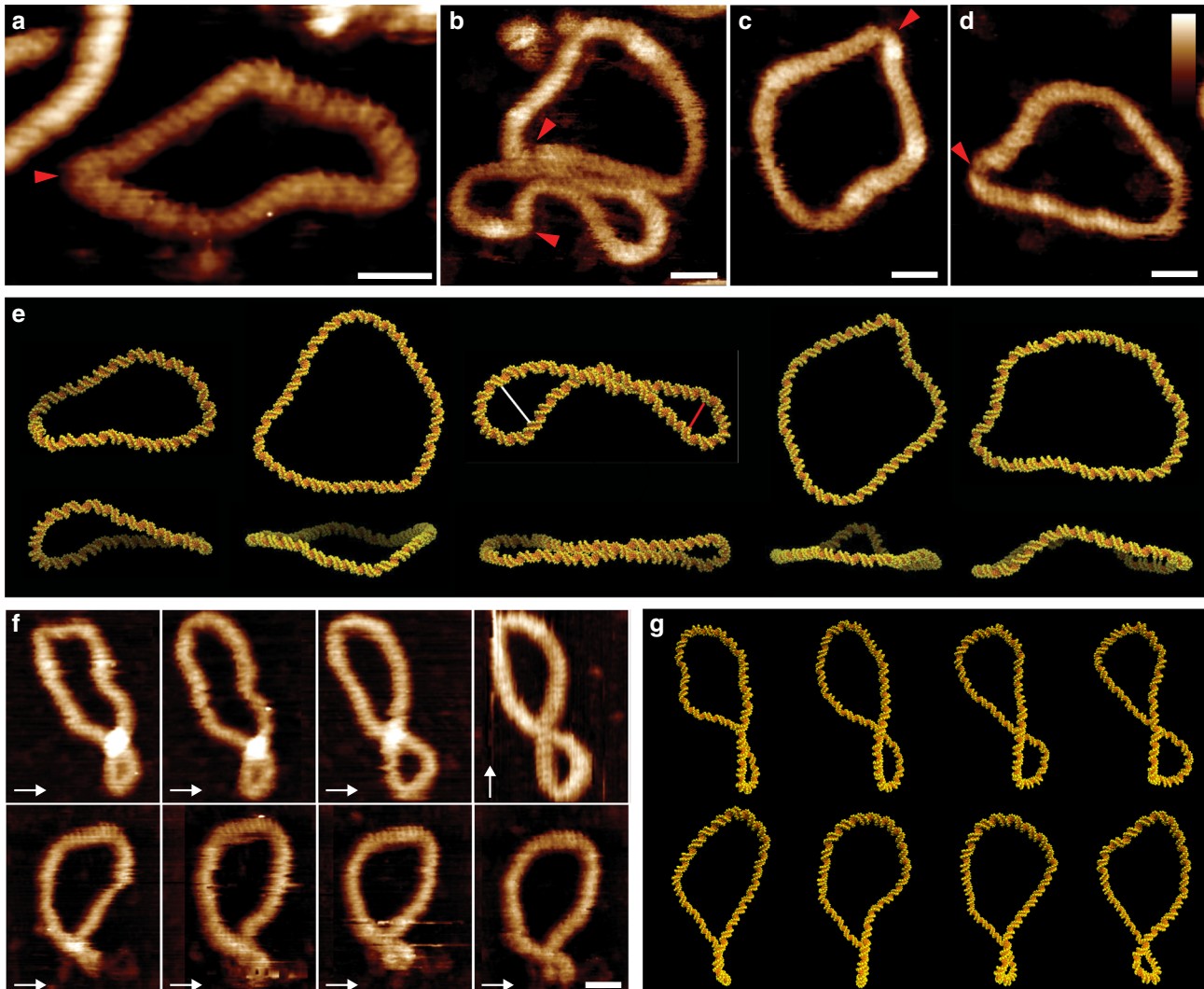

**Fig. 1 Structural and dynamic diversity in supercoiled DNA minicircles. a–d** High-resolution AFM images of natively supercoiled ($\sigma = 0.03–0.06$) DNA minicircles of 251 bp (**a**) and 339 bp (**b–d**) showing their helical structure and disruptions of canonical B-form DNA (marked by red arrowheads), where the angle of the helix changes rapidly, or where the DNA appears thinner or disrupted. Aspect ratios for each molecule: 048 (**a**), 0.44 (**b** bottom), 0.87 (**b** top), 0.78 (**c**) and 0.65 (**d**). **e** MD snapshots of minicircle conformations for 251 (first image) and 339 bp corresponding to the minicircles in the AFM images selected by visual inspection from explicitly solvated simulations (first, second and third images at $\Delta$Lk $-1$, 0 and $-2$, respectively) and from implicitly solvated simulations (fourth and fifth image) at $\Delta$Lk = 0. Top and side views (top and bottom row, respectively) show the degree of planarity of the depicted structures, where top refers to the top view of adsorbed DNA minicircles, and side the perpendicular plane. White and red lines indicate plectonemic loops of 9 and 6.5 nm width, respectively (see 'Methods'). Aspect ratios are 0.45 ± 0.04, 0.30 ± 0.03, 0.86 ± 0.01, 0.81 ± 0.01 and 0.69 ± 0.01. **f** Time-lapse AFM measurements of a natively supercoiled 339 bp DNA minicircle, recorded at 3 min/frame. Fast scan direction is shown by white arrows. **g** Chronological snapshots from simulations of 500 ps duration for a 339 bp minicircle with $\Delta$Lk = $-1$ (see Supplementary Videos 6 and 7). Scale bars (inset): 10 nm and height scale (inset, **d**): 2.5 nm for all AFM images.

the 336 bp minicircles studied by cryo-electron tomography, enzymatic probes detected large defects in negatively supercoiled topoisomers ($\Delta$Lk = $-2$, $-3$ and $-6$), and in highly positively supercoiled DNA ($\Delta$Lk = $+3$). Minor disruptions only were found for $\Delta$Lk $+2$ and $-1$ topoisomers. Our results are entirely consistent with these previous observations (see Fig. 2a, d).

Direct comparison of the level of negative supercoiling required to induce the onset of structural transitions, including denaturation of unbent DNA (typically taken to be around $\sigma \approx -0.04$[36]) with that of DNA minicircles is not straightforward, because the DNA supercoiling response is so exquisitely sequence-dependent. For longer sequences, the statistical likelihood that a sequence will contain an element that undergoes a specific stress-induced structural transition (e.g. Z-DNA formation or cruciform extrusion) is larger[37], and these elements

suppress defect formation by absorbing superhelical stress[38]. Our minicircle sequences do not contain any such supercoiling-responsive sequences. The defects we observe in minicircles are smaller than those that have been probed in 2–5 kbp negatively supercoiled plasmids (>30 bp)[37]. However, coarse-grained simulations of 600 bp supercoiled linear DNA show the formation of small defects (2–3 bp) at plectonemic loops, with larger bubbles (up to 20 bp) observed when plectoneme formation is prohibited by an applied force[39]. We deduce from these observations that DNA bending promotes and localises supercoiling-induced defect formation. Bent-DNA structures are ubiquitous in the genome; as well as forming ends of plectonemes[40], bent DNA is vital to a number of recognition processes, including transcription regulation via DNA looping[41], and DNA damage detection[42,43].

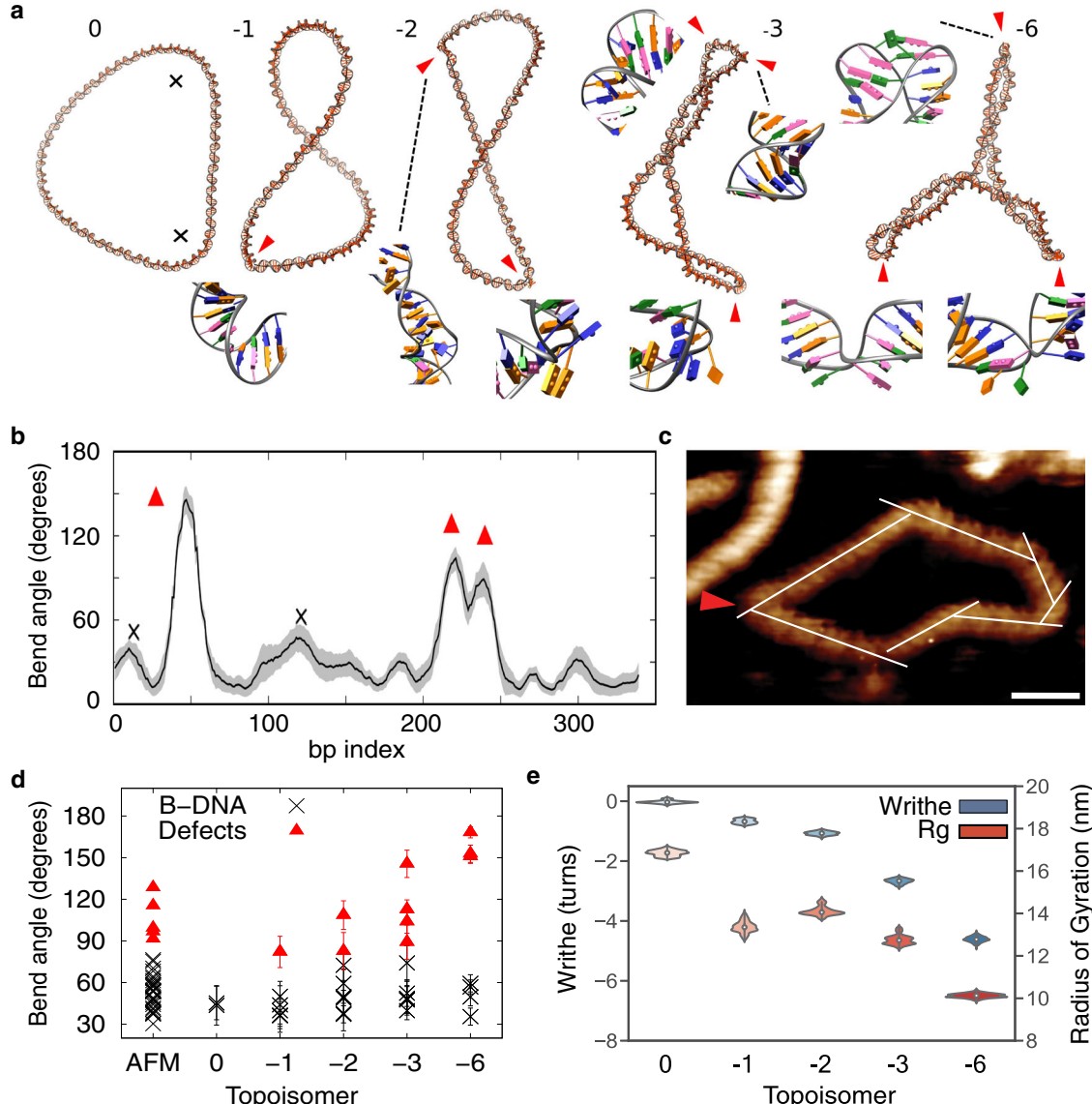

**Fig. 2 Supercoiling induces defect formation in 339 bp DNA minicircles, while increasing writhe and compaction. a** MD average structures showing increased defect formation at higher supercoiling, the numbers at the top of each figure are ΔLk for each structure. **b** Bending calculation obtained by the SerraLINE program using the WrLINE profile from the −3 topoisomer trajectory, where bend angles are calculated as a directional change in tangent vectors separated by 16 bp (additional bending profiles in Supplementary Fig. 3). All peaks >35° are classified as B-DNA bends (black cross) or defects (red triangles) depending on whether canonical non-bonded interactions were broken. **c** Determination of bending angles in natively supercoiled DNA by high-resolution AFM (white lines), scale bar: 10 nm and height scale 2.5 nm. **d** Bent-DNA analysis of DNA minicircles by high-resolution AFM (natively supercoiled, first column), and MD simulations (topoisomers 0 to −6, **a**) shows $a \approx 75°$ cut-off between B-DNA (black crosses) and defects (red triangles), with an increase of the latter with supercoiling. **e** Radius of gyration (Rg) and writhe for the different topoisomers extracted from MD simulations. Grey shading (**b**) corresponds to standard deviations.

**Estimate of critical bend angle associated with defect formation**. We determined the critical bending angle required to form a defect through curvature analysis for all in silico topoisomers (Fig. 2a, b) and for natively supercoiled DNA minicircles by high-resolution AFM (Fig. 2c). Kinks were observed by AFM as discontinuities in the helical repeat of DNA where the angle of the helix changes rapidly, or where the DNA appears thinner or disrupted (Fig. 2c). Defects in the MD were classified as disruptions to base stacking and complementary base pairing (Fig. 2a, insets). Figure 2d shows DNA minicircle bend angles classified as either B-form (black crosses) or defective DNA (red triangles), both for AFM (first column) and MD (all other columns). We deduce that canonical B-form DNA can sustain an angle of up to ~75° on an arc length of approximately one and a half DNA turns

(16 bp for MD, 5 nm for AFM, see 'Methods'), through regions of high bending stress (critical angles of 76° and 74° for AFM and MD, respectively—Supplementary Fig. 3) without disruption to either base stacking or hydrogen bonding. For defective DNA, an average bend angle of $106 \pm 15°$ was measured for AFM and $120 \pm 32°$ for MD, almost double the bend angle measured for canonical DNA, of $69 \pm 5°$ for AFM and $57 \pm 9°$ for MD (mean ± standard deviation). This maximum bend angle of 75° implies that for a DNA bend (such as a plectoneme), to remain free of defects the loop must be >7–10 nm wide, which requires ~55 bp or five helical turns, showing remarkable similarity with coarse-grained simulations[39]. Moreover, it is broadly consistent with the observation that relaxed 63 bp minicircles contain sufficient bending stress that they undergo slow enzymatic digestion when probed

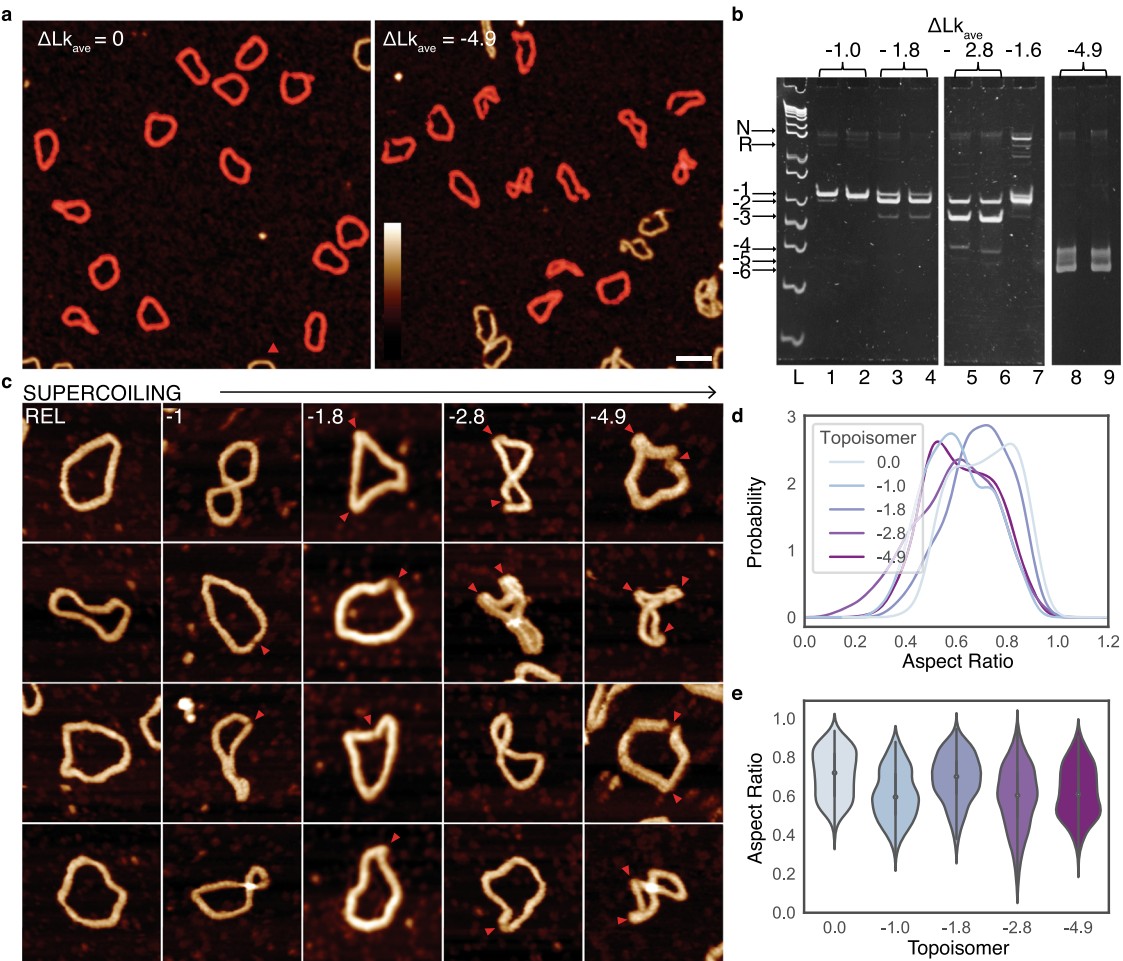

**Fig. 3 Negative supercoiling induces global compaction of DNA minicircles, with a conformational change observed at physiological levels of supercoiling.** **a** AFM images of DNA minicircle populations show increased writhe and compaction at increased negative superhelical density. Images are processed to obtain individual minicircles (red) for analysis[60]. Height scale (inset): 4 nm and scale bar: 50 nm. **b** Five percent TAC acrylamide gel of negatively supercoiled topoisomers of 339 bp ($\Delta$Lk from $-1$ to $-4.9$) generated by the addition of increasing amounts of ethidium bromide during the re-ligation reaction. $\Delta$Lk = $-4.9$ is taken from a separate gel image. N = nicked minicircle; R = relaxed minicircle; markers (left-hand lane) are low molecular weight DNA ladder from NEB (sizes from bottom are: 25, 50, 75, 100, 150, 200, 250, 300, 350 and 500 bp). **c** Representative images of 339 bp minicircles for a range of superhelical densities showing increased levels of compaction and defects (observed as regions of high bending angle, or discontinuities in DNA structure, marked by red arrowheads) for highly supercoiled minicircles. Height scale (inset, **a**): 4 nm and all images are 80 nm wide. **d** The relationship between minicircle aspect ratio and supercoiling as a Kernel Density Estimate (KDE) plot of the probability distribution for each topoisomer ($N$ = 1375). **e** The relationship between minicircle aspect ratio and supercoiling shown as a violin plot for each minicircle topoisomer.

for single-stranded DNA, indicating the presence of minor defects[35] (e.g. type 1 kinks[30]). In the future, continued improvements in other biophysical tools such as Förster resonance energy transfer should reveal further details of the size and flexibility of supercoiling-induced DNA defects and denaturation bubbles, without the necessity for surface immobilisation[44].

**Global compaction in DNA structure correlates with the formation of defects.** To probe how the supercoiling-induced changes in DNA structure vary with the global conformation of DNA minicircles, we generated a range of relaxed and negatively supercoiled topoisomers experimentally (Fig. 3a, b) for comparison with those generated in silico (Fig. 2). For each topoisomer, we quantified the degree of molecular compaction observed by AFM (Fig. 3a) and determined the supercoiling as an average of all bands observed by gel electrophoresis (Fig. 3b). Aspect ratios were calculated for individual minicircles within images (Fig. 3a). While relaxed DNA minicircles appear predominantly as open rings, with high aspect ratio, increasing superhelical density

increases the global compaction generating a range of heterogeneous structures containing defects (Fig. 3c, d). This global compaction from relaxed to maximally supercoiled structures is accompanied by a decrease in the aspect ratio of 35% by AFM (Fig. 3e) and 40% by MD (Fig. 2e).

As expected, as $\Delta$Lk decreases from 0 to $-1$ ($\sigma \approx 0$ to $-0.03$) the DNA writhes and compacts. However, further negative supercoiling of the helix to $\Delta$Lk $\approx -2$ ($\sigma \approx -0.06$) results in a counterintuitive decrease in compaction (Fig. 3e). This correlates with a smaller electrophoretic shift for $-1$ to $-2$ than for the other topoisomer transitions (Fig. 3b) and a smaller change in writhe in the MD simulations than for other transitions ($\Delta$Wr = $-0.4$ turns and $-1.7$ turns for the $-1$ to $-2$ and $-2$ to $-3$ transitions, respectively). This anomalous behaviour correlates with the onset of larger defects observed by both AFM and MD as observed in Figs. 1 and 2. These defects relieve torsional stress and allow the DNA to partially relax, resulting in an increased number of open conformations (Fig. 3c). Comparing the writhe of a defect-containing ($-1.1 \pm 0.1$ turns) and defect-free

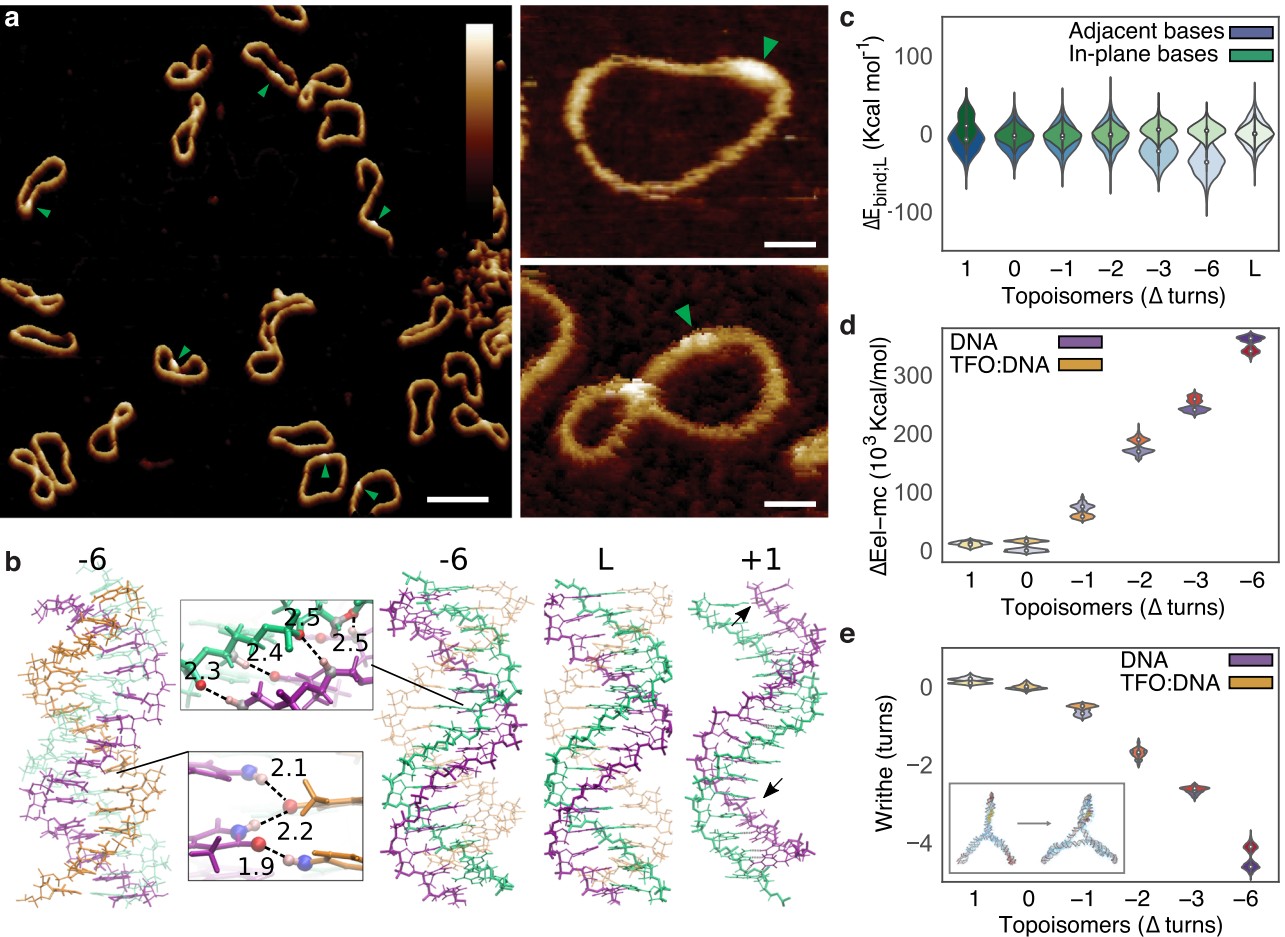

**Fig. 4 Conformational diversity in supercoiled DNA minicircles contributes to the triplex formation. a** AFM images showing triplex formation across a range of DNA minicircle conformations. Triplex regions are visible as small, sub-nanometre protrusions from the DNA marked by green arrowheads. Height scales (scale bar inset): 3 nm and scale bars: (single minicircles) 10 nm and (population): 50 nm. **b** Representative structures of DNA triplex from −6 and +1 topoisomer simulations compared to linear DNA. Arrows indicate less favourable Hoogsteen hydrogen bonds in positively supercoiled DNA. The WC-pyrimidine strand is erased from $\Delta Lk = +1$ image for visualisation purposes. **c** Violin plot of non-bonded interactions for the triplex-binding site ($\Delta E_{bind;L}$), showing the relative contributions from in-plane base interactions (e.g. WC and Hoogsteen hydrogen bonds) (green), compared to interactions between adjacent bases (e.g. bifurcated and backbone hydrogen bonds and stacking energies) (blue). **d** Violin plot for electrostatics of the whole minicircle ($\Delta E_{elec;O}$) with (orange) and without (purple) TFO bound. **e** Minicircle writhe for modelled topoisomers with (orange) and without (purple) TFO bound. Inset shows a half helical turn reduction in writhe on triplex binding for the $\Delta Lk = -6$ topoisomer.

simulation ($-1.7 \pm 0.1$ turns) of the $\Delta Lk = -2$ topoisomer shows that defects cause a reduction in writhe of 0.6 turns (conformers shown in Supplementary Fig. 3). When further negative supercoiling is introduced, the DNA becomes increasingly writhed and compacted, as the superhelical stress can no longer be dissipated purely through the formation of defects.

**Supercoiling-induced conformational variability accommodates binding of TFOs.** The effect of supercoiling-induced structural variability on DNA-binding interactions was investigated through the site-specific binding of a TFO to supercoiled DNA minicircles. The formation of triplex DNA occurs via Hoogsteen base-pairing between the $(CT)_{16}$ TFO and the double-stranded minicircle sequence $(GA)_{16}$[45]. By AFM, we observe triplex formation as small, sub-nanometre protrusions from natively supercoiled DNA minicircles (Fig. 4a). This was verified by AFM measurements on linearised DNA minicircles (Supplementary Fig. 4). Optimisation of the experimental conditions for triplex binding showed that 100 mM divalent (e.g. $Ca^{2+}$ ions) provided the best electrostatic environment (Supplementary Figure 5). Strikingly, surface plasmon resonance (SPR)

experiments showed that the superhelical density of the minicircles has a minimal effect on the affinity of triplex formation, with binding constants ($K_d$) of the order of 10 pM across superhelical densities ($\Delta Lk = 0$ to $-4.9$, Supplementary Fig. 6 and Supplementary Table 1).

To understand the unexpected lack of sensitivity of triplex binding to supercoiling in minicircles, we determined the local and global energetic contributions associated with the binding of the TFO in silico for a range of supercoiled topoisomers (Fig. 4b). Simulation conditions were chosen to mimic the optimal electrostatic environment for triplex binding as determined by SPR (Supplementary Fig. 5). As expected, the relative contributions of the hydrogen bonding and stacking (Fig. 4c) and the local electrostatic (Fig. 4d) interactions vary with superhelical density. As the DNA minicircles are compacted by superhelical stress, the electrostatic penalty for triplex binding increases (Fig. 4d), due to the increase in local negative charge. Taken alone, this would imply that triplex formation is disfavoured by DNA supercoiling; however, the increased electrostatic penalty is offset by the formation of new hydrogen bonds upon triplex formation (Fig. 4c). The new hydrogen bonds (Fig. 4b, inset), preferentially

observed in topoisomers of higher negative supercoiling, consist of (i) bifurcated hydrogen bonds between the Watson–Crick (WC) binding pyrimidine strand and the TFO and (ii) weak hydrogen bonds between the backbone of the WC binding purine strand and the bases of the TFO (Fig. 4c, blue). In addition, negative supercoiling predisposes DNA to the triplex formation, because twist values as low as 30° are observed in triplexes, so triplex formation relieves supercoiling by local unwinding. Conversely, positive supercoiling disrupts Hoogsteen H-bonds disfavouring triplex interaction (Fig. 4c, green).

Local changes in non-bonded interactions with the TFO induce only a minimal perturbation to the mean value of the writhe for all topoisomers apart from $\Delta Lk = -6$. The highly compacted $\Delta Lk = -6$ structure shows a reduction in writhe of $0.5 \pm 0.1$ turns (Fig. 4e) on triplex binding, resulting in a significant shift in the distribution towards more open conformations (structures shown as inset, and in full in Supplementary Fig. 7), presumably due to the electrostatic repulsion associated with these high levels of writhe. For the other topoisomers, triplex binding results in a narrowing of the writhe distribution indicating conformational restriction. This demonstrates that in supercoiled DNA minicircles, global changes in structure and dynamics can be induced by a TFO spanning only one and a half helical turns (16 bp). A balance between the inherent ability of supercoiled minicircles to adopt highly diverse global conformations, and the energetic compensation from the competition of unfavourable electrostatics with increased hydrogen bonding implies that triplex formation should only be minimally affected by the supercoiling-induced variation in the global structure, as is indeed observed by SPR.

## Discussion

Using a combination of high-resolution AFM and atomistic MD simulations, we describe the structure, dynamics and major groove recognition of negatively supercoiled minicircle DNA by TFOs, with double-helical resolution. We quantify the critical bend angle for canonical B-form DNA under superhelical stress as 75°, implying that a DNA loop must be formed of at least five helical turns to be free of defects. These defects dominate DNA mechanics by contributing to the flexibility and conformational diversity of supercoiled DNA.

We observe that superhelical stress globally compacts DNA, resulting in a decreased aspect ratio and radius of gyration. However, at superhelical densities close to that of genomic DNA, we see an unexpected reduction in compaction. We attribute this reduction to the onset of supercoiling-induced type II kinks and denaturation bubbles, through which torsional stress can be dissipated. Beyond this point, the trend to compaction continues, as the defects generated are not sufficient to absorb increased superhelical stress.

The conformational diversity of supercoiled DNA allows for structural perturbations that can accommodate the binding of external substrates, as exemplified by the formation of triplex DNA. The supercoiling dependence of triplex formation is governed by a balance of two competing energetic interactions. An increased electrostatic penalty is incurred in negatively super-coiled DNA due to supercoiling-induced compaction, while additional hydrogen bonds are facilitated by DNA under-twisting. This balance in the energetics facilitates triplex formation across a range of superhelical densities. Our simulations imply that supercoiling in minicircles elevates DNA from its free energy minimum that defines the canonical, linear form onto a relatively flat free energy landscape where multiple conformations become accessible (e.g. writhed or open). We hypothesise that this adaptability of supercoiled DNA, which occurs in part due to the formation of highly flexible denatured regions, increases the

diversity of potential recognition sites. DNA supercoiling provides a molecular mechanism for information at the length scale of one and a half helical turns (e.g. a TFO) to be amplified. Either the range of accessible writhe conformations is affected, or there is a global shift in conformation, as is the case for $\Delta Lk = -6$, where the global writhe changes by 0.5 helical turns when the TFO is present. This suggests that supercoiling can modulate the response of DNA during molecular recognition. Moreover, cryo-electron microscopy imaging[33] and simulations[32] have both observed that kinks and defects can occur co-operatively in minicircles, demonstrating how long-range information transfer in DNA can be facilitated by the imposition of topological constraints. Here, the co-operativity occurs between the global minicircle topology and molecular recognition of a short stretch (16 bp) of the DNA major groove.

Our multiscale simulation protocols combining implicit and explicit solvent allow us to sample a large ensemble of conformations across the six topoisomers. We were thereby able to identify conformers with aspect ratios on average within 7% of the experimental values for all high-resolution AFM images (Fig. 1). Both the superhelical density required to induce denaturation in the 339 minicircles (observed to occur at $\Delta Lk$ −2 both by AFM and MD) and the threshold bend angle for defects (75° over a 5 nm arc length for AFM, 16 bp arc length for simulation) provide a quantitative measure of the ability of MD simulations to reproduce the experimentally observed response of DNA to torsional and bending stress. High-quality AFM images can only be obtained with extremely clean minicircle samples, which can be technically challenging to produce[46]. However, such idealised systems are essential for direct comparisons between simulations and experiments to be valid. We show the remarkable synergy between atomistic simulations and experimental data; despite caveats in both, including a requirement for surface binding, and the use of empirical classical forcefields with sampling limitations imposed by finite computational resources. However, integrating these two biophysical tools enables us to determine the effect of supercoiling on local and global DNA structure and its wider influence on dynamics and recognition. When this is additionally combined with previous biochemical analysis[35], microscopy studies[13,33], theoretical modelling[47] and computer simulations[30–32,39], it is clear that a consensus understanding of the mechanics of small DNA circles is emerging. Although we note that the bending stress for DNA minicircles is much higher than for longer DNA, such as plasmids, or eukaryotic topologically associated domains, the significant perturbation on the mechanics of the DNA enforced by the bending energy in minicircles of this size has particular relevance for the structure of tightly constrained DNA, for example, at plectoneme ends[23,48], short DNA loops and in small extrachromosomal circular DNAs[24,49]. We believe that these data, taken in conjunction with studies of longer DNAs under superhelical stress, will provide a more complete study of DNA structure under stress and can be used to inform future studies on DNA nanotechnology, plectoneme[48] and topology prediction[23]. As well as improving our fundamental understanding of DNA mechanics, our findings have applications in bioengineering, given the proposed therapeutic potential of small circular DNAs and TFOs[50] and the required optimisation of DNA for diagnostics[51] and therapeutics[52].

## Methods

**Generation and purification of small DNA circles**. Small DNA circles (minicircles) of 339 and 251 bp were prepared using bacteriophage λ-Int site-specific recombination in vivo, based on a method previously described with some minor modifications[46] (sequences described in Supplementary information). In each case, a 16-bp triplex-binding site (TBS) (for the triplex-forming oligo TFO1R: 5′[Bt]-CTC TCT CTC TCT CTC T (where Bt indicates biotin), the reverse of the sequence described previously[16]. The primers used in the formation of these small DNA circles are shown in Supplementary Table 2.

Plasmids containing the original minicircle sequences were provided by Lynn Zechiedrich (Baylor College, Houston, TX). For the 251 bp circles, we experienced low yields for the methods described above, thus most materials were obtained from Twister Biotech (Houston, TX, USA); we also obtained larger quantities of 339 bp circles from this company.

The triplex-forming regions were incorporated into the parent plasmids by site-directed mutagenesis using the QuikChange Site-Directed Mutagenesis Kit (Stratagene) following the manufacturer's protocol. Plasmids were transformed into *Escherichia coli* LZ54[46]; the 339 bp minicircles were prepared and isolated using three methods.

For the small-scale (2 L) cultures, a modified version of the protocol developed by Fogg et al.[46] was followed. First, a single colony of *E. coli* LZ54 strain, transformed with the relevant recombination substrate, was used to inoculate 20 mL of LB medium, containing 100 $\mu$g mL$^{-1}$ ampicillin. This was allowed to grow overnight at 30 °C in a standing culture. The overnight culture was next used to inoculate $2 \times 1$ L LB containing 100 $\mu$g mL$^{-1}$ ampicillin in shaker flasks. These, in turn, were grown overnight at 30° C under constant shaking. Cells were harvested by centrifugation under sterile conditions and were resuspended in 50 mL LB. This was used to inoculate 2 L of modified terrific broth medium with 100 $\mu$g mL$^{-1}$ ampicillin. The modified Terrific Broth contained 12 g tryptone, 48 g yeast extract, 30 mL glycerol, 0.1 mL antifoam 204 (Sigma-Aldrich), 2.32 g KH$_2$PO$_4$ and 12.54 g K$_2$HPO$_4$ per litre. Cells were grown at 30° C, while the pH was maintained at 7.0 by the addition of 5% (v/v) phosphoric acid when needed. The dissolved oxygen concentration was maintained at >40% by agitation control. At mid-exponential phase ($A_{600} = 3.5$), Int expression was induced by shifting the temperature to 42° C for 30 min. Norfloxacin was next added to 30 $\mu$g mL$^{-1}$ in order to prevent decatenation by topoisomerase IV, and the temperature was reduced to 30° C, to deactivate Int. After 1 h at 30° C, the cells were harvested by centrifugation. It is worth pointing out that *Bam*HI (which linearises the large circular product that is catenated to the minicircle) was not used to release the minicircle; we found that treatment with *Bam*HI did not increase the yield of the minicircle product. (We presume that the action of DNA topoisomerase IV during cell harvesting was sufficient to achieve this.)

On a larger scale (up to 100 L), 2-L cultures (as described above) were used to inoculate 100 L of modified Terrific Broth in a bioreactor at the Wolfson Fermentation and Bioenergy Laboratory (University of East Anglia, Norwich, UK). The modified Terrific Broth contained 12 g tryptone, 48 g yeast extract, 30 mL glycerol, 0.1 mL antifoam 204 (Sigma-Aldrich), 2.32 g KH$_2$PO$_4$ and 12.54 g K$_2$HPO$_4$ per litre; ampicillin was added to a final concentration of 100 $\mu$g mL$^{-1}$. Cells were grown at 30 °C and the pH was maintained at 7.0 during growth by the addition of 5% (v/v) phosphoric acid when needed. The dissolved oxygen concentration was maintained at >40% by agitation control. Cells were grown to mid-exponential phase ($A_{600} = 3.5$) at which point Int expression was induced by shifting the cultures to 42 °C for 30 min. Norfloxacin (Sigma-Aldrich) was then added to 30 $\mu$g mL$^{-1}$ and the cultures were shifted back to 30 °C. After 1 h, the cells were harvested by centrifugation and the pellet split into ten batches (180 g per batch); the protocol below describes the procedure carried out for each of the cell pellet batches.

The cell pellet was resuspended in 500 mL of 25 mM Tris-HCl (pH 8.0), 50 mM glucose, 10 mM EDTA, and was incubated at room temperature with 2.5 mg mL$^{-1}$ lysozyme (Sigma-Aldrich, chicken egg white) for 30 min. The cells were then lysed by the addition of 1 L 1% sodium dodecyl sulfate, 0.2 M NaOH for 5 min at room temperature, after which 750 mL of 3 M potassium acetate was added. Protein precipitation was allowed to occur for >1 h at 4 °C. Cell debris was removed by centrifugation, and the supernatant was filtered through miracloth under vacuum. Nucleic acid was next precipitated by the addition of isopropanol (0.7 vol) to the filtrate. The resulting harvested pellet was resuspended in 120 mL 10 mM Tris-HCl [pH 8.0], 1 mM EDTA and an equal volume of 5 M LiCl added to precipitate high molecular weight RNA, which was removed by centrifugation. The supernatant was precipitated with ethanol, air dried, resuspended in 150 mL 50 mM MOPS pH 7.0, 5 mM EDTA and then treated with RNase A (Sigma-Aldrich, 50 $\mu$g mL$^{-1}$) for 30 min at 37 °C, followed by proteinase K (Sigma-Aldrich, 50 $\mu$g mL$^{-1}$) for a further 30 min at the same temperature. Most of the unwanted large circle was removed by polyethylene glycol (PEG) precipitation; to the DNA suspension, 150 mL of 10% PEG-8000, 1.5 M NaCl was added and the resulting mixture was incubated at 4 °C for 15 min. The mixture was centrifuged, and the supernatant was treated with 200 mL anion-exchange loading buffer (50 mM MOPS [pH 7.0], 750 mM NaCl, 5 mM EDTA) to reduce the PEG concentration. The DNA minicircles were isolated on QIAGEN-tip 10000 anion-exchange columns following the manufacturer's guidelines. The isolated minicircle was then subjected to Sephacryl S-500 gel filtration to further purify it. Fractions containing minicircle DNA were pooled, and concentrated by isopropanol precipitation, washing the precipitate with ethanol. Purification by gel filtration was repeated a few successive times in order to ensure complete removal of dimeric minicircle. The purified and concentrated minicircle DNA was resuspended in TE buffer.

**Preparation and analysis of different topological species of minicircles**. To generate negatively supercoiled species, the 339 bp minicircle was first nicked at a single site using Nb.BbvCI (New England Biolabs) at 37 °C. After incubation at 80 °C for 20 min to inactivate the endonuclease, nicked DNA was purified and

isolated using the QIAGEN Miniprep Kit. Then, 15 $\mu$g of the purified nicked minicircle was incubated with T4 ligase (New England Biolabs) and ligase buffer containing 25 $\mu$g mL$^{-1}$ bovine serum albumin, in the presence of different quantities of ethidium bromide (EtBr) in a total reaction volume of 3 mL, at room temperature overnight. This was followed by successive purification and isolation of pure supercoiled minicircle DNA using both the QIAGEN nucleotide removal and miniprep kits. The average $\Delta$Lk (linking number difference) for each species was determined by calculating the weighted average of all closed-circular forms by measuring the intensity of each respective band on a polyacrylamide gel (Fig. 3b). The linking number difference ($\Delta$Lk) of each species was assigned by counting bands on gels, as follows: lane 1: $\Delta$Lk$_{ave} = -1.0$; lane 2: $\Delta$Lk$_{ave} = -1.0$; lane 3: $\Delta$Lk$_{ave} = -1.8$; lane 4: $\Delta$Lk$_{ave} = -1.8$; lane 5: native supercoiled ($\Delta$Lk$_{ave} = -1.6$); lane 6: $\Delta$Lk$_{ave} = -2.8$; lane 7: $\Delta$Lk$_{ave} = -2.8$; lane 8: $\Delta$Lk$_{ave} = -4.9$; lane 9: $\Delta$Lk$_{ave} = -4.9$; L = 1 kbp plus ladder (Thermo Fisher Scientific).

Linear forms were prepared by digestion with restriction enzyme *Nde*I (New England Biolabs); relaxed forms were generated either using wheat-germ topoisomerase I (Promega) or by the nicking/ligation procedure described above in the absence of EtBr. DNA samples were analysed by electrophoresis through 5% polyacrylamide gels (acrylamide/bis = 29:1) in TAC (40 mM Tris-acetate [pH 8.0], 10 mM CaCl$_2$) or TAE (40 mM Tris-acetate [pH 8.0], 1 mM EDTA) at 100 V for ~3 h. Gels were stained with SYBR Gold (Invitrogen) and analysed using a Molecular Dynamics STORM 840 Imaging System with quantitation using the ImageQuant software.

Plasmid pBR322 was supplied by Inspiralis Ltd (Norwich, UK) and analysed by electrophoresis through 1% agarose gels in TAE buffer at 80 V for ~2 h. Gels were stained with EtBr and analysed using a Molecular Dynamics STORM 840 Imaging System with quantitation using ImageQuant.

**S1 nuclease digestions**. To determine whether triplex formation between TFO1R and minicircle DNA had occurred, samples were probed with S1 nuclease. To prepare the triplex complex, an excess of TFO1R (2.5 $\mu$M) was incubated with the minicircle/plasmid (150 nM) in 100 mM calcium acetate pH 4.8, in a total volume of 20 $\mu$L at room temperature for 30 min. (In control experiments, reactions were also carried out in TF buffer: 50 mM sodium acetate pH 5.0, 50 mM NaCl, 50 mM MgCl$_2$.) Aliquots (5 $\mu$L) were taken and S1 nuclease (0–1000 U; Thermo Fisher Scientific) was then added and the incubation continued in S1 nuclease buffer (30 mM sodium acetate pH 4.6, 1 mM zinc acetate, 50% [v/v] glycerol) at room temperature for 30 min; the total volume of these reactions was 10 $\mu$L. The digest was stopped by the addition of 0.25 M EDTA (5 $\mu$L), followed by heat inactivation at 70 °C for 10 min; DNA was isolated by extraction with chloroform:isoamyl alcohol.

**DNA minicircle sample preparation for AFM imaging**. Preparation of samples for imaging was carried out as described fully in a published protocol[53]. DNA minicircles were adsorbed onto freshly cleaved mica specimen disks (diameter 3 mm, Agar Scientific, UK) at room temperature, using either Ni$^{2+}$ divalent cations or poly-L-lysine (PLL)[54]. For immobilisation using Ni$^{2+}$, 10 $\mu$L of 20 mM HEPES, 3 mM NiCl$_2$, pH 7.4 solution was added to a freshly cleaved mica disk. Approximately 2 ng of DNA minicircles was added to the solution and adsorbed for 30 min. To remove any unbound DNA, the sample was washed four times using the same buffer solution. For immobilisation using PLL, 10 $\mu$L PLL (0.01% solution, MW 150,000–300,00; Sigma-Aldrich) was deposited on the mica substrate and adsorbed for 1 min. The PLL surface was washed in a stream of MilliQ® ultrapure water, resistivity >18.2 M$\Omega$, and then washed four times with a 50 mM NaOAc pH 5.3 buffer solution to remove any PLL in solution. The supernatant was then removed and 10 $\mu$L 50 mM NaOAc pH 5.3 buffer solution was deposited on the surface. Approximately 2 ng of DNA minicircles was added to the solution and adsorbed for 30 min, followed by four washes in the same buffer to remove any unbound DNA minicircles. Ni$^{2+}$ immobilisation was used to obtain the data shown in Figs. 1 and 2 and PLL for the data shown in Fig. 3.

**Triplex containing DNA minicircle sample preparation for AFM imaging**. For experiments with TFO, DNA minicircles were incubated in an Eppendorf with a tenfold excess of TFO in 50 mM NaOAc buffer at pH 5.3, prior to adsorption onto the mica substrate using the PLL method, as above. To verify the location of the TFO on the DNA sequence (Supplementary Fig. 4), the minicircles were first linearised by cutting with *Nde*I.

**AFM imaging**. All AFM measurements were performed in liquid following a published protocol[53]. All experiments except Fig. 1f were carried out in PeakForce Tapping imaging on Multimode 8 and FastScan Bio AFM systems (Bruker). In these experiments, continuous force–distance curves were recorded with the tip–sample feedback set by the peak force as referenced to the force baseline. The following cantilevers were used: MSNL-E (Bruker) Peakforce HiResB (Bruker) and biolever mini (Olympus, Japan) on the Multimode 8, and FastScan D (Bruker) on the FastScan Bio with approximately equal resolution obtained by each. Force–distance curves were recorded over 20 nm (PeakForce Tapping amplitude of 10 nm), at frequencies of 4 kHz (Multimode 8) and 8 kHz (FastScan Bio). Imaging

was carried out at PeakForce setpoints in the range of 5–20 mV, corresponding to peak forces of <70 pN. Images were recorded at 512 × 512 pixels to ensure a resolution ≥1 nm/pixel at line rates of 1–4 Hz.

Figure 1f was obtained on a home-built microscope with a Closed-Loop PicoCube XYZ Piezo Scanner (PhysikInstrumente, Karlsruhe, Germany) and with a Fabry–Perot interferometer to detect the cantilever deflection[20]. FastScan D (Bruker) cantilevers were actuated photothermally in tapping mode at amplitudes of 1–2 nm. Imaging was carried out at line rates of 3 Hz, over scan sizes of 50 nm with a setpoint ~80% of the free amplitude. Imaging forces are extremely difficult to calculate in tapping mode[55–57], can be quite sensitive to ambiguity in the measurement of the reference 'free' amplitude used, and can drift substantially from those initially set. To avoid such difficulties, imaging forces were estimated by observing the compression of the DNA compression of the molecule, with the average height for each molecule calculated to be 1.5 ± 0.03 nm ($N = 7$, mean ± s.d.), which correlates to a peak force of ~100 pN[22] (Supplementary Fig. 8).

**AFM image processing**. The methods used for automated processing and tracing of DNA are described fully here[58], with the code available at https://github.com/AFM-SPM/TopoStats [59]. Here, AFM images were processed using a user-designed Python script (pygwytracing.py), which utilises the Gwyddion 'pygwy' module[60] for automated image correction, DNA molecule identification and morphological analysis. The algorithm searches recursively for files within a user-defined directory. This search also excludes any files of the format '_cs', which are cropped files exported by the Nanoscope Analysis software (Bruker, CA, USA). AFM images are loaded using gwyddion functions and topography data are automatically selected using the choosechannels function. The pixel size and dimensions of each image are determined using the imagedetails function, which allows all inputs to be specified in real, that is, nanometre values, in place of pixel values. This is especially important for datasets with changing resolution.

Basic image processing is performed in the function editfile, which uses the functions: 'align rows' to remove offsets between scan lines; 'level' to remove sample tilt as a first-order polynomial; 'flatten base', which uses a combination of facet and polynomial levelling with automated masking; and 'zeromean', which sets the mean value of the image, that is, the background, to zero. A gaussian filter ($\sigma = 1.5$) of 3.5 pixels (1–2 nm) was applied to remove pixel errors and high-frequency noise.

Single DNA molecules are identified in images using a modified extension of Gwyddion's automated masking protocols, in which masks are used to define the positions of individual features (grains) on the imaged surface. The grains within a flattened AFM image are identified using the 'mask_outliers' function, which masks data points with height values that deviate from the mean by >1$\sigma$ (with 3$\sigma$ corresponding to a standard gaussian). Grains that touch the edge of the image (i.e. are incomplete) are removed using the 'grains_remove_touching_border' function and grains that are <200 nm$^2$ are removed using the 'grains_remove_by_size' function. Erroneous grains are removed using the removelargeobjects and removesmallobjects functions, which themselves use the function 'find_median_pixel_area' to determine the size range of objects to remove. The 'grains_remove_by_size' function is then called again to remove grains, which fall outside 50–150% of the median grain area determined in the previous step.

Grain statistics are then calculated for each image using the 'grainanalysis' function, which utilises the 'grains_get_values' function to obtain a number of statistical properties, which are saved using the saveindividualstats function as '.json' and '.txt' files for later use in a subdirectory 'GrainStatistics' in the specified path. In addition, each grain's values are appended to an array [appended_data], to statistically analyse the morphologies of DNA molecules from all images for a given experiment (presumed to be within a single directory). This array is converted to a pandas dataframe[61] using the 'getdataforallfiles' function and saved out using the savestats function as '.json' and '.txt' files with the name of the directory in the original path.

Individual grains (i.e. isolated molecules) are cropped out using the function bbox, which uses the grain centre $x$ and $y$ positions obtained in the 'grainanalysis' function to duplicate the original image and crop it to a predefined size (here 80 nm) around the centre of the grain. These images are then labelled with the grain ID and saved out as tiff files in a subdirectory 'Cropped' in the specified path.

To allow for further processing in python, there is an option to obtain the image or mask as a numpy array[62], using the function 'exportasnparray'. The processed image and a copy with the mask overlaid are saved out using the 'savefiles' function to a subdirectory 'Processed' in the specified path.

Statistical analysis and plotting are performed using the 'statsplotting' script. This script uses the 'importfromjson' function to import the JSON format file exported by 'pygwytracing' and calculates various statistical parameters for all grain quantities, for example, length and width, and saves these out as a new JSON file using the 'savestats' function. Both Kernel Density Estimate plots and histograms are generated for any of the grain quantities using the matplotlib[63] and seaborn[64] libraries within the functions 'plotkde', 'plotcolumns' and 'plothist'.

**Determination of minicircle bend angles by AFM**. To determine the bend angles for DNA minicircles by AFM, images were imported into Gwyddion[60], and basic processing was carried out as described above in the 'editfile' script for basic flattening. Bend angles were then measured between straight parts ≥5 nm using

Gwyddion's measurement tool, achieving thus a resolution of approximately one DNA turn and a half.

**Determination of triplex binding by AFM**. To verify that the small protrusions observed on DNA in the presence of the TFO at low pH were triplexes, the site of the protrusions was determined. The 339 bp minicircles were linearised at the NdeI site and imaged by AFM as described above. Processed images were traced by hand in IMOD[65] (University of Colorado, CO, USA) to determine the position of the protrusion along the DNA (Supplementary Fig. 4). The tracing data were analysed using the TFOlength script. The mean and standard deviation for each length measurement (full minicircle, triplex and triplex flanking lengths) were calculated using built-in functions, and the data for each plotted as a histogram.

The length of the minicircle was determined as 109 ± 4 nm, with the triplex measured as 37 ± 2 nm, 34% of the length of the minicircle. The distance between the TFO site and the restriction site is 127 bp, which is 37% of the length of the minicircle, and in good agreement with the AFM measurements. The length of the triplex as measured by AFM is 6 ± 2 nm. Errors quoted are standard deviations.

**Atomistic simulations: set up of the structures for supercoiled 339 bp DNA minicircles**. Linear starting DNA molecules with the same 339 bp sequence as above were built using the NAB module implemented in AmberTools12[66]. DNA planar circles corresponding to six topoisomers ($\Delta Lk = -6, -3, -2, -1, 0, 1$) with/without the 16 bp triplex-forming oligomer were constructed using an in-house program. The AMBER99 forcefield[67] with different corrections for backbone dihedral angles including the parmBSC0 for $\alpha$ and $\gamma$[68], the parmOL4 for $\chi$ (glycosidic bond)[69] and the parmOL1 for $\varepsilon$ and $\zeta$[70] were used to describe the DNA. These forcefield improvements correct known artefacts such as the underestimate of the equilibrium twist of DNA, and biases in $\varepsilon$ and $\zeta$ torsion angles, which may have generated non-physical conformers in previous minicircle simulations[31]. Parameters for protonated cytosine present in the triplex-forming oligomer were obtained from Soliva et al.[71]. Following our standard protocol[72], the SANDER module within AMBER12 was used to subject the starting structures for the different types of minicircles to 20 ns of implicitly solvated MD using the Generalised Born/Solvent Accessible area method[73] at 300 K and 200 mM salt concentration, with the long-range electrostatic cut-off set to 100 Å. Restraints were imposed on the complementary (e.g. WC) hydrogen bonds between paired DNA bases. Due to the neglect of solvent damping, the timescales in implicitly solvated MD are accelerated relative to simulations performed in the solvent by at least tenfold[13].

**Simulations of 339 bp minicircles in explicit solvent**. To select the starting structure for explicitly solvated simulations, we performed clustering analysis using the average linkage algorithm within PTRAJ for the implicitly solvated DNA trajectories. Representative structures of the most populated clusters then were chosen and solvated in TIP3P rectangular boxes with a 6 nm buffer, 339 Ca$^{2+}$ counterions[74] to balance the DNA charge and additional Ca$^{2+}$/2Cl$^-$ ion pairs[75] corresponding to 100 mM. These specific simulation conditions were chosen to mimic the optimal electrostatic environment observed for triplex binding by SPR (see Supplementary Fig. 5). Two replicas of the −2 and −3 topoisomers were subjected to 100 ns explicitly solvated MD simulations, starting from the two most representative structures. Single 100 ns MD simulations were performed for topoisomers −6, −1, 0 and +1. Solvated MD runs were performed using the GROMACS 4.5 program[76] with standard MD protocols[72] at 308 K and, afterwards, were carefully visualised to ensure that rotation of the solute was not significantly compared to the size of the simulation box over the timescale of the MD. Only the last 30 ns sampled every 10 ps were used for the subsequent analysis. VMD[77] and Chimera[78] were used to depict representative structures, to measure the longest distance across plectonemic loops and to detect defective DNA through visual inspection. DNA defects were confirmed through energetic analysis of stacking and hydrogen bonds at the relevant base steps using GROMACS 4.5. Hydrogen bonds were determined using 3.5 Å and 140° as a distance and angle cut-off, respectively, as in Fig. 4d.

Additional simulations for $\Delta Lk = -6, -2$ and 0 topoisomers were performed using BSC1 forcefield corrections[79] for DNA backbone dihedral angles instead of parmOL4. The BSC1 forcefield has been designed to correct previous artefacts while simultaneously maintaining the generality of the forcefield[79]. Simulations were started using the same initial structures and were run with equivalent solvent conditions in TIP3P rectangular boxes with a 3 nm buffer for 100 ns using CUDA version of AMBER16[79]. Again, trajectories were carefully visualised to ensure that rotation of the solute was not significant compared to the size of the simulation box over the timescale of the MD[80]. Only the last 30 ns and a snapshot every 10 ps were used for the subsequent analysis, which are presented in Supplementary Fig. 9

**Simulations of linear DNA in explicit solvent**. A 36-mer fragment containing the TBS was extracted from the 339 bp minicircle to compare binding energies of this site on unconstrained linear DNA or on supercoiled minicircles. The TBS was placed in the middle to avoid end effects[81]. The linear starting structure was solvated explicitly for running MD simulations and was set up, minimised and equilibrated following the protocols described previously.

**Simulations of 260 bp minicircles**. The structure used for mirroring the high-resolution AFM image of a 251 bp minicircle on Fig. 1a was extracted from a simulation previously run for the −1 topoisomer of a 260 bp minicircle[72]. The slightly longer sequence of 260 bp was constructed based on the experimental sequence of 251 bp studied here.

**Global shape and other geometrical analysis of simulations**. The radius of gyration was determined using the AMBER program PTRAJ[82]. Other geometrical descriptions of the global shape, such as writhe and bend, were performed using the WrLINE molecular contour[83] and SerraLINE programs (both software suites are freely accessible at https://github.com/agnesnoy). With SerraLINE, the bending angles $\theta$ were calculated from the directional correlation, $\theta = \cos^{-1}(z_i \cdot z_j)$, where $z_i$ and $z_j$ are the two tangent vectors. Each $z_i$ was obtained by combining two successive points of the WrLINE global contour $(z_i = r_{i+1} - r_i)$. Bending angles $\theta$ were calculated using two tangent vectors ($z_i$ and $z_j$) separated by 16 nucleotides (approximately a DNA helical turn and a half) as a compromise length for capturing the overall bend produced by a defect or by canonical B-DNA. The bending profiles in Supplementary Fig. 3 were obtained by scanning all the possible 16-bp sub-fragments along the minicircle, and the peaks over 35° were selected to compare the MD simulations with the AFM data shown in Fig. 2d. Following these criteria, we obtained a total of 23 B-DNA bends and 10 kinks. SerraLINE was also used to calculate the degree of planarity through the minimal perpendicular distances between the WrLINE molecular contours and best-fitting planes for each individual frame of simulations. Aspect ratios were then obtained via the longest and shortest axes of the molecular contours projected to the above-calculated plane, thus mirroring the Gwyddion software method used for AFM image analysis.

**Selection of MD conformers for visual comparison with AFM structures**. Our multiscale simulation protocols combining implicit and explicit solvent sample a large ensemble of conformations because the supercoiled minicircles are extremely flexible. Implicit solvent simulations can explore global structural parameters such as the writhe (Supplementary Fig. 2). Explicitly solvated calculations provide a more limited set of conformers over MD timescales because of the high viscosity of the solvent; however, it is only with this more accurate description that we can observe the formation of kinks and denaturation bubbles at the local base-pair level, which in turn leads to a compaction of the DNA (Fig. 2e). For comparison with the five high-resolution 2D AFM images of natively supercoiled minicircles (which have an average superhelical density of 0.05), we visually inspected a total of 3000 (explicitly solvated) and 1000 (implicitly solvated) frames from topoisomers in the range $\Delta Lk = -2$ to 0. Explicit simulations at $\Delta Lk$ −1, 0 and −2 with aspect ratios of $0.45 \pm 0.04$, $0.30 \pm 0.03$ and $0.86 \pm 0.01$ were matched to AFM structures with aspect ratios of 048 (Fig. 1a), 0.44 (Fig. 1b, bottom) and 0.87 (Fig. 1b, top). Conformers from implicit solvent simulations at $\Delta Lk = 0$ with aspect ratios of $0.81 \pm 0.01$ and $0.69 \pm 0.01$ were matched to minicircles (Fig. 1c, d) with aspect ratios of 0.78 (Fig. 1c) and 0.65 (Fig. 1d). Although the aspect ratios have been used here as a measure of the structural similarity, the implicit solvent simulations are unable to observe the denaturation bubbles seen in the AFM images.

**Energy calculations of triplex DNA formation**. To obtain theoretical insight into the thermodynamics driving triplex binding, we used the MD trajectories to estimate the global (e.g. electrostatics) and local (e.g. base-pair stacking and hydrogen bonding) contributions to the overall binding energy. The global electrostatic contribution for configurational energy ($E_{ele}$) was evaluated using the AMBER program MMPBSA[84]. To compare between the different topoisomers, the individual components were referred to the relaxed DNA-naked topoisomer ($\Delta E_{elec; 0} = E_{elec} - E_{elec;0}$) as it is shown in Fig. 4d.

We also analysed the interaction energy between nucleotides in the triplex binding site, considering in-plane base interactions and nearest neighbours only (e.g. 9 bases in total). All interaction energies were calculated using the GROMACS 4.5 program. The two components of the binding energy were calculated at the TBS for each topoisomer by discarding the effect of the unbound third strand ($E_{bind} = E_{TRI} - E_{DNA}$). Values for the different topoisomers were referenced to the linear fragment ($\Delta E_{bind; L} = E_{bind} - E_{bind;L}$), as shown in Fig. 4c. The local interaction energy terms (Lennard–Jones and electrostatic interactions) between in-plane nucleotides were used as an estimate of the Hoogsteen hydrogen bonding at the TBS (Fig. 4c, green), while the interaction energies between bases in the planes above and below were used as an estimation of base stacking, hydrogen bonding and non-bonded backbone interactions (Fig. 4d, blue). The presence of these hydrogen bonds was confirmed by visual inspection in VMD (Fig. 4b).

**Surface plasmon resonance**. SPR measurements were recorded at either 25 °C or 35 °C using a Biacore T200 system (GE Healthcare). All experiments were performed using an SA Series S Sensor Chip (GE Healthcare), which has four flow cells each containing streptavidin pre-immobilised to a carboxymethylated dextran matrix. For immobilisation, a standard immobilisation protocol was used with a running buffer of HSB-EP+ buffer (10 mM HEPES pH 7.4, 150 mM NaCl, 3 mM EDTA, 0.05% [v/v] surfactant P20). The chip surface was first washed using three injections of 1.0 M NaCl, 50 mM NaOH for 60 s, each followed by buffer for 60 s (all at 10 μL min$^{-1}$). The 5′-biotinylated TFO (TFO1R, 30–60 nM) was then immobilised onto two of the flow cells (FC2 and FC4) and a response of ~250 response units (RUs) was aimed for. The remaining two flow cells (FC1 and FC3) were kept free of ligand and were used as reference cells.

Experiments were carried out using conditions that were modified from those reported previously with plasmid pNO1[16], optimised for the 339 minicircle. Using these optimised conditions (TFO1R [250 RU immobilised], flow rate 2 μL min$^{-1}$, 100 mM calcium acetate pH 4.8, 25 °C, injection time 600 s; regeneration: 1 M NaCl, 5 mM NaOH, 60 s, 30 μL min$^{-1}$) a range of differently supercoiled samples at 50 nM of 339 nr ($\Delta Lk$ −4.9 to +1), as well as relaxed, nicked and linear, and samples containing no triplex-forming sequences were injected and the binding monitored.

The kinetics of the binding between the small-circle DNA substrate and the TFO1R ligand were then measured using a multi-cycle kinetics approach using the same optimised conditions but with 221 RU of TFO1R immobilised. For the kinetic experiments, 339 bp minicircles ($\Delta Lk$ −4.9, −2.8, linear and relaxed) were injected over flow cells 1 and 2 for 600 s at a range of concentrations (2.5, 5, 10, 20, 30 and 40 nM) and a buffer-only control. A buffer-only solution was then flowed for 1 h so that the dissociation could be more accurately recorded. The SA chip was regenerated after each injection of DNA using 1 M NaCl, 5 mM NaOH. The experiment was carried out at 35 °C with a flow rate of 2 μL min$^{-1}$ using 100 mM calcium acetate pH 4.8 as the running buffer. The inclusion of buffer-only controls enabled the use of double referencing, whereby, for each analyte measurement, in addition to subtracting the response in the reference flow cells from the response in the test flow cells, a further buffer-only subtraction was made to correct for the bulk refractive index changes or machine effects[85]. The data were analysed using the Biacore T200 Evaluation software version 2.0 using the kinetics fit assuming a 1:1 binding model.

**Statistics and reproducibility**. For AFM, sample sizes were based on the reproducibility of the result and on previous experience. Each experiment was repeated multiple times and showed the same trend when analysed using automated code. For high-resolution measurements (Fig. 1), less repeats were obtained due to the difficulty of these measurements; however, multiple molecules from at least two samples were analysed. AFM replicas were performed successfully with the same results over a period of 4 years by two separate co-authors. For AFM analysis of the effect of supercoiling on the overall structure of DNA minicircles, only data taken using the same immobilisation and imaging method was used. This was to ensure results were comparable across multiple datasets. Images were only excluded for AFM if the data quality was too poor to allow the data to be automatically processed, and therefore allowed for consistent exclusion reducing bias. For MD simulations, 3000 frames were taken from last 30 ns of each simulation every 10 ps for subsequent analysis. Tests were done using 30,000 values or the last 20 ns with no significant difference. The only exception is Fig. 2d where B-DNA bends stronger than 30° (in total 23) and all kinks (10) were used. Note each bend value was obtained following the previous rules. Two MD replicas were run for the −2 and the −3 topoisomers successfully being presented in the current study, MD replicas were successful.

**Reporting summary**. Further information on research design is available in the Nature Research Reporting Summary linked to this article.

## Data availability
Data supporting the findings of this manuscript are available from the corresponding authors upon reasonable request. A reporting summary for this Article is available as a Supplementary information file.

The atomic force microscopy data and atomistic molecular dynamics simulation data generated and/or analysed during the current study are available in the figshare repository https://doi.org/10.6084/m9.figshare.13116890 [86].

## Code availability
All code written and used in this study is available via github, with AFM analysis scripts at: https://github.com/AFM-SPM/TopoStats [58,59] and MD writhe line scripts at: https://github.com/agnesnoy/SerraLINE [87].

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

## Acknowledgements

We thank Lynn Zechiedrich and Jonathan Fogg for informative conversations and for supplying minicircle plasmids, the quality of which made these experiments possible; Andrea Slade, James Shaw, Bede Pittenger, Shuiqing Hu, Chanmin Su and Thomas Mueller (Bruker) for assistance in developing equipment and protocols for high-resolution imaging of DNA; Maya Topf, Agnel Praveen Joseph, Luzie Helfmann, Robert Gray and Christopher Soelistyo for assistance with developing the automated AFM analysis; Andrea Hall for assistance with fermentation; and Charlie Laughton for the program to build in silico DNA minicircles with triplex DNA. This work was supported by grants BB/I019294/1 and BB/J004561/1 from the Biotechnology and Biological Sciences Research Council (BBSRC), by grants EP/M506448/1, EP/M028100/1, EP/N027639/1, and EP/R513143/1 from the Engineering and Physical Sciences Research Council (EPSRC), by a UKRI/MRC Rutherford Innovation fellowship MR/R024871 from the Medical Research Council (MRC) and by the John Innes Foundation. Time on ARCHER and JADE was granted via the UK High-End Computing Consortium for Biomolecular Simulation, HECBioSim, supported by the EPSRC (EP/R029407/1). We also acknowledge the use of the Leeds Advanced Research Computing service.

## Author contributions

A.L.B.P., A.N., B.W.H., A.D.B, A.M. and S.A.H. conceived and designed the experiments. A.L.B.P. and K.H.S.M. conducted AFM experiments, A.L.B.P. and J.G.B. wrote software to analyse AFM images and performed analysis. A.N. performed and analysed MD simulations and V.V.-B. developed SerraLINE program. F.M.C., M.M.P. and C.E.M.S. performed triplex-binding SPR experiments. A.L.B.P., L.A.M., F.M.C. and M.M.P performed gel electrophoresis and prepared the different samples. A.L.B.P., A.N. and S.A.H analysed the data and wrote the manuscript with input from B.W.H, A.D.B. and A.M.

## Competing interests

The authors declare no competing interests.
