## [Peer Review File · Nature Communications]

Reviewer #1 (Remarks to the Author):

Pyne and coworkers present a joint simulation/experimental study on how supercoiling affects DNA conformation and its ability to bind to oligonucleotides. They use a combination of AFM and MD to characterize and quantify the frequency of defects in the DNA helix, which lead to flexible hinges that can partly relieve twists in the DNA. They show examples from MD and AFM that are in qualitative agreement, to motivate their combined use to probe the molecular nature and prevalence of these supercoil-induced defects. This includes understanding characteristic geometric features (R_g , writhe), especially the observation of sharp kinks indicative of the aforementioned hinges. They show the surprising result that triplex-forming oligonucleotides are not significantly affected by the presence of the supercoiled structure, which the authors attribute to a balance of an increase in the electrostatic binding penalty with a concomitant increase in the number of hydrogen bonds. The authors use simulation to demonstrate this interplay.

Overall, this is an interesting study, with implications on the behavior of supercoiled DNA structures and their interaction with surrounding species. While I am on the fence if the results are broad/general enough to warrant publication in Nature Communications, this should be published with some revisions.

Major revisions/clarifications:

-Not a big change, but I think an important one...the authors portray their results as more general than they really are. Particularly, the title of the article gives the impression that the results will provide some rather sweeping arguments v/v DNA-binding species, but the focus here is on TFOs. I think that is fine - and appropriate given that this is a vast area of inquiry - but this should be reflected in the title, intro, and conclusions.

-The authors should be clear in Fig. 1 if these are representative snapshots from simulations, how they were able to obtain structures that are so close to those in the AFM observations. What is the relative prevalence of similar structures, i.e. is there a way to quantify how representative these simulations are? Approximately how many structures were observed in simulation versus AFM experiments? Please provide some context.

-The authors justify the matching with AFM based on the planarity of the DNA structures. I find this unconvincing, since these are not rigid planar structures. It seems likely that the chains could still be distorted by the surface, especially given the statement, "...demonstrating that dynamic behavior can occur in these molecules on the order of minutes..." In disagreement with the statement that immediately follows, "...implying that the structural perturbation caused by immobilization is small.", this slow relaxation process suggests that the surface strongly interacts with the adsorbed DNA. The act of adsorbing to such a 'sticky' surface would likely perturb its conformation; for example, it seems implausible that the chains would be spontaneously oriented so that they lay gently on this surface, and if one part of the chain adsorbs then the rest of the chain would follow - perhaps with enough force as to promote the formation of defects.

So, I do not see any evidence in the manuscript that adsorption would not drastically change the DNA conformation. I think it is reasonable to perform the comparison, though, because there is not an alternative method with similar resolution. I ask the authors to more explicitly acknowledge these limitations, and present further validation of these results as an area for further inquiry.

-The argument for the minimal effect of superhelical density on TFO binding is reasonable, but based solely on simulation data. Given that one of the contributions - the electrostatic penalty for triplex binding - can be tuned via salt concentration, is it possible to test this computational observation directly in experiment? At the very least, please suggest how this hypothesized competition could be tested in experiment.

Minor revisions:

-The introduction of 'linking number' in the introduction is abrupt for the broad readership of Nat. Comm. I recommend providing a brief definition of this and related (i.e. writhe and twist) quantities.

-The use of a bar graph in Fig. 2e is odd. Why not use a scatterplot like 2d?

Reviewer #2 (Remarks to the Author):

Pyne et al. show beautiful AFM images of DNA mini-circles and compare those with structures they observe in MD simulations. They use this to study the effect of supercoiling on defects in the B-form of the DNA. The argumentation seems solid. The use of high-resolution AFM imaging and MD simulation is very promising for understanding details in molecular conformations. It is nice to see the two images side by side. It would have been nice, however, if the authors had made more use of this combination and really tried to match the helical structures they observe in the AFM with the structures they get from MD simulation. Technologically the manuscript is solid, although some claims go farther than I believe is substantiated by the data, see comments listed below.

Comments :

1) Page 3 : "Fig 1a-e is attributed to thermal fluctuations within supercoiled DNA, with time-resolved AFM (Fig. 1f) demonstrating that dynamic behaviour can occur in these molecules on the order of minutes, even when tethered to a surface, also implying that the structural perturbation caused by immobilisation is small. Similar dynamics were observed in MD simulations of the 339 minicircle ($\Delta Lk = -1$) in a continuum representation of the solvent, albeit at a much faster (picosecond) rate (Fig. 1g and Supp. Videos 1, 2). «

This is misleading, since it suggests that the AFM image corresponds to what the MD simulation shows. I can hardly believe this. There are too many unknowns and red-flags that don't allow to draw that conclusion. It is not reasonable to assume that ALL the mica does is slow down the dynamics by a factor of $1E10$! If the mica affects the timescale of the process that strongly, then one would assume that it also affects the geometric pathway of the process. The authors also don't seem to take into account the additional energy that they add to the system with the AFM tip. This will also drive the system in different configurations. I, therefore, would suggest the authors remove this statement and comment more on the difficulties. This wouldn't diminish the value of the paper, but give the reader sufficient information to put the results into perspective.

2) Comments to figure 1f: I have several concerns about this image:

- It is striking that the image quality of this figure is less good than that of the other images. This is presumable because they performed these measurements with a different machine, and not in

PeakForce tapping but in conventional tapping mode. I do wonder why though? The images of figure 1f were recorded at 3 lines per second, whereas the PeakForce tapping images were recorded at line rates from 1-4 Hz, and gotten higher quality images? So presumably the authors could have taken these images also with the Dimension FastScan, or am I mistaken? Is there another reason to use the other microscope?

- The authors should clearly indicate where one image stops, and one image starts.

- the 4th timepoint (top right) appears to have a different “noise pattern”, almost as if this image was scanned top to bottom rather than left to right? Do the authors have an explanation for this?

- It is not clear to me how the authors can compare the “peak force” in their PeakForce tapping images on the Bruker Multimode and FastScan to the “peak force” in the time-resolved measurements. How do they convert from their tapping mode parameters to the peak force? For this calculation, it is insufficient to state the tapping mode amplitude as 1-2nm, since it makes a big difference if it is 1nm or 2. The setpoint of 80% also seems very high. Other groups (Ando et. al) image at 90-95% setpoint?

- The paper cited for the force estimate is not appropriate. In reference 19, the measurements were done in phase-feedback, not in tapping mode as was done in this manuscript. Estimating the tip-sample forces in high-speed tapping mode is very difficult, as has been shown by several publications (Xu et al: 10.1529/biophysj.108.132829, Nievergelt et al: DOI : 10.1038/s41565-018-0149-4, Kielar et al: Angew. Chem. Int. Ed. 10.1002/anie.202005884). I therefore very much doubt the 100pN value mentioned in the methods section.

- The red triangles are not well suited as it isn't clear where they point. The authors should replace them with an arrow or at least arrowheads.

Reviewer #3 (Remarks to the Author):

The strongest aspect of this MS consists of amazingly good AFM images of DNA minicircles. On these images, readers can clearly see individual strands of DNA and even distinguish major and minor DNA grooves. I have been working in DNA microscopy field for over 40 years and the shown AFM images are the best I saw up to now. The second strongest point is the perfect correspondence between the images of DNA minicircles and numerical simulations of the same minicircles. This perfect correspondence convinces readers that at the scale as small as DNA base pairs, the authors and also readers can understand what happens when DNA minicircles are underwound to the level that is close to natural DNA supercoiling.

The major results of the paper are not original in a strict sense as these aspects of DNA mechanics were investigated over many years by many groups. However, this study makes a quantum jump by directly visualizing the studied phenomena at the scale of individual turns of the double helix. This study will certainly have a great effect on thinking in the field.

I ask the authors to consider the following points during the revision:

1. Page 3. The authors wrote that the planarity of the minicircles was high as it was less than 15%. If some measure is high then its value should not be less than some value but bigger than some value. Presumably the authors wanted to say that non-planarity of the minicircles was low and it was less than 15%.

2. Page 4. Legend to Fig. 1. The authors wrote about top and side views of simulated DNA

minicircles. In my opinion, in simulations there are no top or side views as these directions are not defined. One could accept expressions “top” and “side” views but still better would be to say about directions corresponding to top and side views of adsorbed DNA minicircles.

3. Page 5. The authors wrote that they deduced from their observations that DNA bending promotes and localizes supercoiling-induced defect formation. This statement gives an impression that the authors are first to demonstrate this effect but this is not the case. The authors need to add that they confirmed several earlier studies of supercoiling induced DNA bending in DNA minicircles such as: Du et al. (2008) Kinking the double helix by bending deformation. *Nucleic Acids Res.*, 36, 1120–1128. Mitchell et al. (2011) Atomistic simulations reveal bubbles, kinks and wrinkles in supercoiled DNA. *Nucleic Acids Res.*, 39, 3928–3938. Zheng, X. and Vologodskii, A. (2009) Theoretical analysis of disruptions in DNA minicircles. *Biophys. J.*, 96, 1341–1349. Lionberger et al. (2011) Cooperative kinking at distant sites in mechanically stressed DNA. *Nucleic Acids Res.*, 39, 9820-9832.

4. Page 5 and also later in the text the authors discuss the critical bending angle of DNA helix but do not specify over which arc length of DNA this bending angle is measured. Without this specification the presented values are not really relevant. Only on page 19, I could find the information that the arc length over which the bending was measured was 16 bp. This information should be provided much earlier.

5. Page 9. The title of the section at the top of the page is misleading as in fact the binding of triplex forming oligonucleotides was hardly affected by supercoiling.

Reviewer #4 (Remarks to the Author):

The study by Pyne, Harris and co-workers applies state of the art molecular dynamics simulation and atomic force microscopy to understand negatively coiled mini-circles of DNA and their interactions with triplex forming oligonucleotides. There is clear agreement between the simulation and experiment and the data is well presented. The only minor concerns are potential force field dependence observed in earlier MD simulations of mini circles which instead of kinking deformed in other ways and if there is a force field dependence; however this is only a minor concern since the simulation and experiment seem to match. Additionally, in the methods section it states that restraints for all hydrogen bonds were applied when likely this was just for Watson-Crick base pair hydrogen bonds.

EDITORIAL COMMENTS:

We have made all of the editorial changes requested by the office, and this has significantly improved the transparency of our data and the clarity of our presentation. We have further improved the manuscript by responding to the reviewer comments below, incorporating their suggestions into the paper, and including additional data (SPR, AFM and MD), as specified in the response below.

REVIEWER COMMENTS

Reviewer #1 (Remarks to the Author):

Pyne and coworkers present a joint simulation/experimental study on how supercoiling affects DNA conformation and its ability to bind to oligonucleotides. They use a combination of AFM and MD to characterize and quantify the frequency of defects in the DNA helix, which lead to flexible hinges that can partly relieve twists in the DNA. They show examples from MD and AFM that are in qualitative agreement, to motivate their combined use to probe the molecular nature and prevalence of these supercoil-induced defects. This includes understanding characteristic geometric features (R_g , writhe), especially the observation of sharp kinks indicative of the aforementioned hinges. They show the surprising result that triplex-forming oligonucleotides are not significantly affected by the presence of the supercoiled structure, which the authors attribute to a balance of an increase in the electrostatic binding penalty with a concomitant increase in the number of hydrogen bonds. The authors use simulation to demonstrate this interplay.

Overall, this is an interesting study, with implications on the behavior of supercoiled DNA structures and their interaction with surrounding species. While I am on the fence if the results are broad/general enough to warrant publication in Nature Communications, this should be published with some revisions.

Major revisions/clarifications:

-Not a big change, but I think an important one...the authors portray their results as more general than they really are. Particularly, the title of the article gives the impression that the results will provide some rather sweeping arguments v/v DNA-binding species, but the focus here is on TFOs. I think that is fine - and appropriate given that this is a vast area of inquiry - but this should be reflected in the title, intro, and conclusions.

The title of the manuscript has been changed, and we have reworded the description of these finding in the Conclusions to indicate that we investigated the recognition of DNA by Triplex Forming Oligos (TFOs) specifically. We agree with this reviewer that the presentation of the findings on molecular recognition in the paper was somewhat narrow. We have clarified that TFOs are used as a model system in the Introduction, where we explicitly explain why we considered this a good choice. In the Conclusions, we have also followed this referee's advice, and commented on the wider implications to recognition of supercoiled DNA substrates to our findings for TFOs.

-The authors should be clear in Fig. 1 if these are representative snapshots from simulations, how they were able to obtain structures that are so close to those in the AFM observations. What is the relative prevalence of similar structures, i.e. is there a way to quantify how representative these simulations are? Approximately how many structures were observed in simulation versus AFM experiments? Please provide some context.

Both the AFM experiments and MD simulations provide 'snapshots' as a representation of the broad distribution of conformations seen within supercoiled DNA. Our multiscale simulation protocols combining implicit and explicit solvent allow us to sample a large ensemble of conformations across the six topoisomers. Given that we are matching 3D atomistic models to a 2D AFM image, we only need to sample structures with approximately the correct writhe to be able to identify MD conformers that visually match an experimental structure. We achieved this in practice by identifying conformers through visual inspection, validating the agreement by comparing the aspect ratios, as we now explain in the text with more detail included in a separate section in the methods. The aspect ratios for each molecule in Figure

1 were determined to be: 0.48, (a), 0.44 (b top), 0.87 (b bottom), 0.78 (c), 0.65 (d) by AFM and 0.45 ± 0.04 , 0.30 ± 0.03 , 0.86 ± 0.01 , 0.81 ± 0.01 , 0.69 ± 0.01 by MD. The methods for calculating these are explained in the text, and can be briefly summarised as: measurement of the minimum divided by maximum bounding length. Most of these molecules were within 10% of each other, showing good agreement between experiment and simulations.

Molecule	AFM (arb. units)	MD (arb. units)
a	0.48	0.45
b (i)	0.44	0.3
b (ii)	0.87	0.86
c	0.78	0.81
d	0.65	0.69

To make our procedure for extracting conformers completely transparent, and to emphasise the large structural diversity observed in our combined implicit/explicit solvent simulations we have included a new supplementary figure (SI Fig. 2 a, b) demonstrating the writhe in the relevant trajectories, and showing where the conformers selected appeared. We have included other, representative configurations in the figure so that readers can appreciate the diversity of minicircle conformers obtained.

-The authors justify the matching with AFM based on the planarity of the DNA structures. I find this unconvincing, since these are not rigid planar structures. It seems likely that the chains could still be distorted by the surface, especially given the statement, "...demonstrating that dynamic behavior can occur in these molecules on the order of minutes..." In disagreement with the statement that immediately follows, "...implying that the structural perturbation caused by immobilization is small.", this slow relaxation process suggests that the surface strongly interacts with the adsorbed DNA. The act of adsorbing to such a 'sticky' surface would likely perturb its conformation; for example, it seems implausible that the chains would be spontaneously oriented so that they lay gently on this surface, and if one part of the chain adsorbs then the rest of the chain would follow - perhaps with enough force as to promote the formation of defects.

So, I do not see any evidence in the manuscript that adsorption would not drastically change the DNA conformation. I think it is reasonable to perform the comparison, though, because there is not an alternative method with similar resolution. I ask the authors to more explicitly acknowledge these limitations, and present further validation of these results as an area for further inquiry.

Though these molecules are formed of flexible DNA, due to their closed covalent form, and short length (twice the persistence length of DNA), they have high degrees of planarity and rigidity. To make the degree of deviation from planarity clearer to the reader, we have now included videos showing 3D rotational views of the structures observed (Supp. Videos 1-5). In addition to these new data, we have also referred the reader to the movies presented by Irobalieva et al.¹ (Supplementary Movies 2 and 3), which shows that a large degree of planarity was also observed in 3D structures of supercoiled minicircles by cryo-ET (albeit at lower resolution than we achieve here). These authors have confirmed the formation of defects in these structures via enzymatic probes which do not require surface immobilisation, in good agreement with the published literature². We have expanded on the use of biochemical assays to probe defect formation in DNA minicircles², and highlighted additional relevant studies in the text³ (pg 6).

-The argument for the minimal effect of superhelical density on TFO binding is reasonable, but based solely on simulation data. Given that one of the contributions - the electrostatic penalty for triplex binding - can be tuned via salt concentration, is it possible to test this computational observation directly in

experiment? At the very least, please suggest how this hypothesized competition could be tested in experiment.

As the reviewer clearly states, the binding of TFO's can be affected by both monovalent and divalent ions, and this can be tested using surface plasmon resonance (SPR). SPR experiments were carried out prior to simulations being run to determine the best binding conditions for triplex formation, and these conditions used in the simulation. We found that the addition of divalent calcium ions increases triplex formation up to a maximum of 100 mM Ca²⁺ (Supp. Fig. 5). We also tested the effect of monovalent ions, using K⁺ and Na⁺ and found that these reduced the efficiency of triplex formation (Supp. Fig. 5). We therefore used 100 mM Calcium Acetate for both simulations, and also for SPR measurements of the affinity of triplex formation in supercoiled DNA minicircles (Supp. Fig. 6). We have added this data to the supplementary information (Supp. Fig. 5), and clarified this in the text (pg 10).

Minor revisions:

-The introduction of 'linking number' in the introduction is abrupt for the broad readership of Nat. Comm. I recommend providing a brief definition of this and related (i.e. writhe and twist) quantities.

We have added an additional description of supercoiling including twist and writhe to the first paragraph.

-The use of a bar graph in Fig. 2e is odd. Why not use a scatterplot like 2d?

We have changed this bar graph to a violin plot to show the full dataset.

Reviewer #2 (Remarks to the Author):

Pyne et al. show beautiful AFM images of DNA mini-circles and compare those with structures they observe in MD simulations. They use this to study the effect of supercoiling on defects in the B-form of the DNA. The argumentation seems solid. The use of high-resolution AFM imaging and MD simulation is very promising for understanding details in molecular conformations. It is nice to see the two images side by side. It would have been nice, however, if the authors had made more use of this combination and really tried to match the helical structures they observe in the AFM with the structures they get from MD simulation. Technologically the manuscript is solid, although some claims go farther than I believe is substantiated by the data, see comments listed below.

The huge diversity of conformations observed in the simulations and in the AFM experiments means that direct matching of structures does not provide a quantitative measure of agreement between the simulations and the experiments, because any structural descriptor we choose pulls out many good matches to any single experimental conformer from the MD, and *vice versa*. We acknowledge that this was not sufficiently clear in the original version of the paper, and have added Supplementary Figure 2, which explicitly shows the diversity of conformations observed by MD, to clarify this vital concept.

The key measurements we make that do, however, directly compare the predictions of the simulations with the experiments is the level of supercoiling and the critical bend angle required to induce defect formation. We have emphasised these points in the manuscript.

In response to the feedback from both yourself and reviewer 1, we have also performed a numerical comparison of the conformers identified through visual inspection in Figure 1. We do this by comparing the aspect ratios, as we now explain in the text with more detail included in a separate section in the methods. The methods for calculating these are explained in the text, and can be briefly summarised as: measurement of the minimum divided by maximum bounding length. Most of these molecules were within 10% of each other, showing good agreement between experiment and simulations (see table above).

Comments:

1) Page 3 : "Fig 1a-e is attributed to thermal fluctuations within supercoiled DNA, with time-resolved AFM (Fig. 1f) demonstrating that dynamic behaviour can occur in these molecules on the order of minutes, even when tethered to a surface, also implying that the structural perturbation caused by immobilisation

is small. Similar dynamics were observed in MD simulations of the 339 minicircle ($L_k = -1$) in a continuum representation of the solvent, albeit at a much faster (picosecond) rate (Fig. 1g and Supp. Videos 1, 2). «

This is misleading, since it suggests that the AFM image corresponds to what the MD simulation shows. I can hardly believe this. There are too many unknowns and red-flags that don't allow to draw that conclusion. It is not reasonable to assume that ALL the mica does is slow down the dynamics by a factor of $1E10$! If the mica affects the timescale of the process that strongly, then one would assume that it also affects the geometric pathway of the process. The authors also don't seem to take into account the additional energy that they add to the system with the AFM tip. This will also drive the system in different configurations. I, therefore, would suggest the authors remove this statement and comment more on the difficulties. This wouldn't diminish the value of the paper, but give the reader sufficient information to put the results into perspective.

We are extremely sorry that our results were presented in such a way that this referee thought we were implying that the MD configurations were sampled in the same order as those observed in the experiments. This is absolutely not the case, because of course both the MD simulations and the conformers explored experimentally are both stochastic and highly conformationally diverse, so agreement between the two would be statistically impossible. We have added a sentence on page 3 to explicitly refer to the energy imparted by the tip during AFM imaging, which may enable the adhered DNA minicircles to explore their free energy landscape. We have also added a sentence on page 3 at the end of the paragraph to refer explicitly to the time series of the MD simulations as being different due to the random statistical nature of thermal fluctuations (Supp. Fig. 2).

2) Comments to figure 1f: I have several concerns about this image: - It is striking that the image quality of this figure is less good than that of the other images. This is presumable because they performed these measurements with a different machine, and not in PeakForce tapping but in conventional tapping mode. I do wonder why though? The images of figure 1f were recorded at 3 lines per second, whereas the PeakForce tapping images were recorded at line rates from 1-4 Hz, and gotten higher quality images? So presumably the authors could have taken these images also with the Dimension FastScan, or am I mistaken? Is there another reason to use the other microscope?

Over the past years, we have used different microscopes to record high-resolution images of DNA.^{4,5} Although we found similar results with different microscopes, there were some differences in the ease and robustness at which these results could be obtained on different microscopes. In brief, the reason for using the other microscope here is purely historical; and yes, similar images could have been taken with the Dimension FastScan.

- The authors should clearly indicate where one image stops, and one image starts.

We have modified Figure 1 to add white separation between images

- the 4th timepoint (top right) appears to have a different "noise pattern", almost as if this image was scanned top to bottom rather than left to right? Do the authors have an explanation for this?

This image was taken in the opposite scan direction to observe the double helical structure along the vertical direction of the molecule, as we obtain better resolution in the fast scan direction. We have added arrows to Figure 1 to highlight the fast scan direction and referred to these in the caption.

- It is not clear to me how the authors can compare the "peak force" in their PeakForce tapping images on the Bruker Multimode and FastScan to the "peak force" in the time-resolved measurements. How do they convert from their tapping mode parameters to the peak force? For this calculation, it is insufficient to state the tapping mode amplitude as 1-2nm, since it makes a big difference if it is 1nm or 2. The setpoint of 80% also seems very high. Other groups (Ando et. al) image at 90-95% setpoint?

Setpoint indications in tapping mode are always relative to the free amplitude "just" above the surface, but with these smaller cantilevers, there is a rather rapid change of the amplitude in the last μm before the tip and sample get into contact, due to hydrodynamic interactions (see Leung et al., Nano Lett 2012)⁴. Hence there can significant ambiguity in the reference that is considered for indicating the relative

setpoint, which is probably the reason for the difference with reported relative setpoints by Ando et al.. In addition, there are questions to what extent tapping mode force estimations are accurate at all (see e.g. Nievergelt et al., Nat Nanotechnol 2018)⁶. To avoid such ambiguity, we estimated the imaging force based on the compression of the DNA, based on previous measurements of the measured DNA height as a function of imaging force (Pyne et al., Small 2014)⁵, which leads to the (effective) tapping mode force estimates indicated in the manuscript. See also the next, related point.

- The paper cited for the force estimate is not appropriate. In reference 19, the measurements were done in phase-feedback, not in tapping mode as was done in this manuscript. Estimating the tip-sample forces in high-speed tapping mode is very difficult, as has been shown by several publications (Xu et al: 10.1529/biophysj.108.132829, Nievergelt et al: DOI : 10.1038/s41565-018-0149-4, Kielar et al: Angew. Chem. Int. Ed. 10.1002/anie.202005884). I therefore very much doubt the 100pN value mentioned in the methods section.

As the reviewer states, it is extremely hard to determine applied force in tapping mode. We apologise for our initial citation, and thank the reviewer for provision of more appropriate citations which we have added to the manuscript. As mentioned in our response above, we estimated our applied force by using the DNA height as a reference, based on past measurements of the DNA height as a function of applied force ⁵, thus avoiding complications in estimating imaging forces in tapping mode. To make this clearer we have added an additional supplementary figure (Supp. Fig. 8) showing the height of the DNA molecules, both for a 'zoomed out' image of the DNA, and for each DNA minicircle in the time series. We have also added the following text to the methods:

Imaging forces are extremely difficult to calculate in tapping mode⁶⁻⁸, can be quite sensitive to ambiguity in measurement of the reference "free" amplitude used, and can drift substantially from those initially set. To avoid such difficulties, imaging forces were estimated by observing the compression of the DNA molecule, with the average height for each molecule calculated to be 1.5 ± 0.03 nm ($N = 7$, mean \pm std), which correlates to a peak force of ~ 100 pN⁴ (Supp. Fig. 8)."

- The red triangles are not well suited as it isn't clear where they point. The authors should replace them with an arrow or at least arrowheads.

Red triangles have been replaced with red arrow heads to point at the location of defects in figures 1, 2 & 3, and supplementary figures 2 & 9. Green triangles have been replaced with green arrow heads in figure 4.

Reviewer #3 (Remarks to the Author):

The strongest aspect of this MS consists of amazingly good AFM images of DNA minicircles. On these images, readers can clearly see individual strands of DNA and even distinguish major and minor DNA grooves. I have been working in DNA microscopy field for over 40 years and the shown AFM images are the best I saw up to now. The second strongest point is the perfect correspondence between the images of DNA minicircles and numerical simulations of the same minicircles. This perfect correspondence convinces readers that at the scale as small as DNA base pairs, the authors and also readers can understand what happens when DNA minicircles are underwound to the level that is close to natural DNA supercoiling.

The major results of the paper are not original in a strict sense as these aspects of DNA mechanics were investigated over many years by many groups. However, this study makes a quantum jump by directly visualizing the studied phenomena at the scale of individual turns of the double helix. This study will certainly have a great effect on thinking in the field.

We thank the reviewer for their positive comments, and have addressed each of the points below in the text.

I ask the authors to consider the following points during the revision:

1. Page 3. The authors wrote that the planarity of the minicircles was high as it was less than 15%. If some measure is high then its value should not be less than some value but bigger than some value.

Presumably the authors wanted to say that non-planarity of the minicircles was low and it was less than 15%.

We have corrected this in the text:

“The deviation from planarity of the minicircles was calculated to be less than 15% on average (Supp. Fig. 1 and Supp. Videos 1-5)”

2. Page 4. Legend to Fig. 1. The authors wrote about top and side views of simulated DNA minicircles. In my opinion, in simulations there are no top or side views as these directions are not defined. One could accept expressions “top” and “side” views but still better would be to say about directions corresponding to top and side views of adsorbed DNA minicircles.

We have corrected this in the text:

“Top and side views (top and bottom row, respectively) show the degree of planarity of the depicted structures, where top refers to the top view of adsorbed DNA minicircles, and side the perpendicular plane.”

3. Page 5. The authors wrote that they deduced from their observations that DNA bending promotes and localizes supercoiling-induced defect formation. This statement gives an impression that the authors are first to demonstrate this effect but this is not the case. The authors need to add that they confirmed several earlier studies of supercoiling induced DNA bending in DNA minicircles such as: Du et al. (2008) Kinking the double helix by bending deformation. *Nucleic Acids Res.*, 36, 1120–1128. Mitchell et al. (2011) Atomistic simulations reveal bubbles, kinks and wrinkles in supercoiled DNA. *Nucleic Acids Res.*, 39, 3928–3938. Zheng, X. and Vologodskii, A. (2009) Theoretical analysis of disruptions in DNA minicircles. *Biophys. J.*, 96, 1341–1349. Lionberger et al. (2011) Cooperative kinking at distant sites in mechanically stressed DNA. *Nucleic Acids Res.*, 39, 9820-9832.

We agree with the reviewer and thank them for the references. We have added these in the text, and emphasised the work already carried out within the field on pages 5 and 6:

“This maximum bend angle of 75°, implies that for a DNA bend (such as a plectoneme), to remain free of defects the loop must be more than 7-10 nm wide, which requires approximately 55 bp or five helical turns, showing remarkable similarity with coarse grained simulations⁹. Moreover, it is broadly consistent with the observation that relaxed 63 bp minicircles contain sufficient bending stress that they undergo slow enzymatic digestion when probed for single stranded DNA, indicating the presence of minor defects² (e.g. type 1 kinks¹⁰). In the future, continued improvements in other biophysical tools such as FRET should reveal further details of the size and flexibility of supercoiling-induced DNA defects and denaturation bubbles, without the necessity for surface immobilisation³.”

4. Page 5 and also later in the text the authors discuss the critical bending angle of DNA helix but do not specify over which arc length of DNA this bending angle is measured. Without this specification the presented values are not really relevant. Only on page 19, I could find the information that the arc length over which the bending was measured was 16 bp. This information should be provided much earlier.

We have now explicitly stated this in the text:

“We deduce that canonical B-form DNA can sustain an angle of up to ~75° on an arc length of approximately one and a half DNA turns (16 bp for MD, 5 nm for AFM)”.

We have also clarified this in the AFM methods section.

5. Page 9. The title of the section at the top of the page is misleading as in fact the binding of triplex forming oligonucleotides was hardly affected by supercoiling.

We have changed the title to better reflect this:

“Supercoiling-induced conformational variability accommodates binding of triplex-forming oligonucleotides”

Reviewer #4 (Remarks to the Author):

The study by Pyne, Harris and co-workers applies state of the art molecular dynamics simulation and atomic force microscopy to understand negatively coiled mini-circles of DNA and their interactions with triplex forming oligonucleotides. There is clear agreement between the simulation and experiment and

the data is well presented. The only minor concerns are potential force field dependence observed in earlier MD simulations of mini circles which instead of kinking deformed in other ways and if there is a force field dependence; however this is only a minor concern since the simulation and experiment seem to match. Additionally, in the methods section it states that restraints for all hydrogen bonds were applied when likely this was just for Watson-Crick base pair hydrogen bonds.

This referee is absolutely correct that our force-fields for describing DNA are continuously improving, and that previous structural changes observed in MD simulations of minicircles may have been artefactual. We have added a sentence in the methods to provide this historical context, we have added a poignant example of this from the literature, and we have explained which issues have been addressed by the subsequent DNA force-field refinements employed in this work. We have included the following in the methods to clarify this:

“Restraints were imposed on the complementary (e.g. Watson-Crick) hydrogen bonds between paired DNA bases” (pg 20)

“These forcefield improvements correct known artefacts such as the underestimate of the equilibrium twist of DNA, and biases in ϵ and ζ torsion angles, which may have generated non-physical conformers in previous minicircle simulations³¹.” (pg 20)

“The BSC1 forcefield has been designed to correct previous artefacts while simultaneously maintaining the generality of the force-field⁷⁶.” (pg 21)

References:

1. Irobalieva, R. N. *et al.* Structural diversity of supercoiled DNA. *Nat. Commun.* **6**, 8440 (2015).
2. Du, Q., Kotlyar, A. & Vologodskii, A. Kinking the double helix by bending deformation. *Nucleic Acids Res.* **36**, 1120–1128 (2008).
3. Craggs, T. D. *et al.* Substrate conformational dynamics facilitate structure-specific recognition of gapped DNA by DNA polymerase. *Nucleic Acids Res.* gkz797 (2019) doi:10.1093/nar/gkz797.
4. Leung, C. *et al.* Atomic Force Microscopy with Nanoscale Cantilevers Resolves Different Structural Conformations of the DNA Double Helix. *Nano Lett.* **12**, 3846–3850 (2012).
5. Pyne, A., Thompson, R., Leung, C., Roy, D. & Hoogenboom, B. W. Single-Molecule Reconstruction of Oligonucleotide Secondary Structure by Atomic Force Microscopy. *Small* **10**, 3257–3261 (2014).
6. Nievergelt, A. P., Banterle, N., Andany, S. H., Gönczy, P. & Fantner, G. E. High-speed photothermal off-resonance atomic force microscopy reveals assembly routes of centriolar scaffold protein SAS-6. *Nat. Nanotechnol.* **13**, 696–701 (2018).
7. Xu, X., Carrasco, C., de Pablo, P. J., Gomez-Herrero, J. & Raman, A. Unmasking Imaging Forces on Soft Biological Samples in Liquids When Using Dynamic Atomic Force Microscopy: A Case Study on Viral Capsids. *Biophys. J.* **95**, 2520–2528 (2008).
8. Kielar, C. *et al.* On the Stability of DNA Origami Nanostructures in Low-Magnesium Buffers. *Angew. Chem. Int. Ed.* **57**, 9470–9474 (2018).
9. Matek, C., Ouldrige, T. E., Doye, J. P. K. & Louis, A. A. Plectoneme tip bubbles: Coupled denaturation and writhing in supercoiled DNA. *Sci. Rep.* **5**, 7655 (2015).
10. Lankaš, F., Lavery, R. & Maddocks, J. H. Kinking Occurs during Molecular Dynamics Simulations of Small DNA Minicircles. *Structure* **14**, 1527–1534 (2006).

Reviewer #1 (Remarks to the Author):

The authors fully addressed my comments, and given the improvements to the manuscript I recommend publication in Nature Communications.

Reviewer #2 (Remarks to the Author):

The authors have suitably adressed my comments and I recommend publication of this very nice manuscript.